



# Water mass transformation variability in the Weddell Sea in Ocean Reanalyses

Shanice Bailey[1], Spencer Jones[3], Ryan Abernathey[1], Arnold Gordon[1], and Xiaojun Yuan[2]

[1]Department of Earth & Environmental Sciences of Columbia University Lamont-Doherty Earth Observatory | Room 106 Geoscience Bldg. P.O. Box 1000 Palisades, NY 10964
[2]Lamont Doherty Earth Observatory | P.O. Box 1000 - 61 Route 9W Palisades, NY 10964
[3]Department of Oceanography at Texas A&M University | Eller O&M Building, College Station, TX 77843

**Correspondence:** Shanice Bailey (stb2145@columbia.edu)

**Abstract.** This study investigates the variability of water mass transformation (WMT) within the Weddell Gyre (WG). The WG serves as a pivotal site for the meridional overturning circulation (MOC) and ocean ventilation because it is the primary origin of the largest volume of water mass in the global ocean, Antarctic Bottom Water (AABW). Recent mooring data suggest substantial seasonal and interannual variability of AABW properties exiting the WG, and studies have linked the variability to

the large-scale climate forcings affecting wind stress in the WG region. However, the specific thermodynamic mechanisms that link variability in surface forcings to variability in water mass transformations and AABW export remain unclear. This study explores WMT variability via WMT volume budgets derived from Walin's classic WMT framework, using three state-of-the-art, data-assimilating ocean reanalyses: Estimating the Circulation and Climate of the Ocean state estimate (ECCOv4), Southern Ocean State Estimate (SOSE) and Simple Ocean Data Assimilation (SODA). From the model outputs, we diagnose a closed

form of the water mass budget for AABW that explicitly accounts for transport across the WG boundary, surface forcing, interior mixing, and numerical mixing. We examine the annual mean climatology of the WMT budget terms, the seasonal climatology, and finally the interannual variability. In ECCO and SOSE, we see strong interannual variability in AABW volume budget. In SOSE, we find an accelerating loss of AABW, driven largely by interior mixing and changes in surface salt fluxes. ECCO shows a similar trend during a 3-yr time period beyond what is covered in SOSE, but also reveals such trends to be part of interannual

variability over a much longer time period. Overall, ECCO provides the most useful timeseries for understanding the processes and mechanisms that drive WMT and export variability. SODA, in contrast, displays unphysically large variability in AABW volume, which we attribute to its data assimilation scheme. We examine correlations between the WMT budgets and large-scale climate indices, including ENSO and SAM; no strong relationships emerge, suggesting that these reanalysis products may not reproduce the AABW export pathways and mechanisms hypothesized from observations.

# 1   Introduction

The Meridional Overturning Circulation (MOC) is an essential component of ocean circulation that has important global effects on the Earth's climate system. The MOC is responsible for transporting and redistributing heat, salt, carbon and nutrients between hemispheres (Purkey and Johnson, 2013; Vernet et al., 2019). It does so by exchanging surface waters with deep waters,





creating a gateway for the abyssal oceans to communicate with the atmosphere. In the Atlantic sector of the ocean, warm,
salty water in the South Atlantic is transported to the north where it loses heat at the surface, sinking into the deep ocean and
forming North Atlantic Deep Water (NADW)(Talley, 2013; Vernet et al., 2019). NADW transports water, heat, carbon and
nutrients south, eventually reaching Antarctica where it upwells to the near surface and undergoes density changes due to air-sea
interaction, transforming into other water masses that are major constituents of forming Antarctic Bottom Water (AABW). The
formation sites of AABW are also locations of major carbon sinks (Grant et al., 2006; Brown et al., 2015). The global impacts
of this circulation system on biological productivity and carbon and heat uptake, particularly in the context of climate change,
make the MOC worth studying extensively.

The Weddell Gyre (WG) is the most prominent gyre in the Southern Ocean. The WG is a clockwise-flowing subpolar gyre
east of the Antarctic Peninsula, driven by the wind stress curl in this region. The WG is a dynamic region where relatively
warmer waters from the north bring in heat; as these waters upwell they are cooled and freshened, transforming into surface
water masses that become part of the sources for bottom water formation. The WG serves as a pivotal site for the MOC and for
ventilating the ocean because it is the primary origin (supplying 40-50 %, Stewart (2021)) of the densest volume of water mass
in the global ocean, Antarctic Bottom Water (AABW).

The lower cell of the MOC is the branch containing AABW. AABW is the coldest and densest water mass in the Southern
Ocean, and comprises about 36 % of the global deep ocean volume (Johnson, 2008; Purkey and Johnson, 2013; Purkey et al.,
2018). The lower cell circulation is dependent upon surface forcing and interior mixing. Turbulent mixing sets the depth to which
AABW upwells, and provides a mechanism for waters to transform into AABW and circulate into the abyssal ocean (Nikurashin
and Vallis (2011), Nikurashin and Vallis (2012), Ferrari et al. (2014), Nycander et al. (2015)). The upper cell is ventilated mainly
through open-ocean subduction. In this process, water masses form in the mixed layer and acquire their characteristics through
air-sea exchanges of heat and dissolved gases, which are transferred down into the thermocline (Williams, 2001). In contrast, the
lower cell is ventilated through coastal polynyas and plumes of dense, high-oxygen-containing cold waters cascading down the
continental boundaries.

The summary above focuses on the mean state of the MOC. However, recent mooring data suggest significant seasonal and
interannual variability of AABW properties exiting the WG (Gordon et al., 2020). It is hypothesized that these variabilities are
linked to the coupling of large-scale climate forcings, the El Niño Southern Oscillation and the Southern Annular Mode (SAM),
through wind stress variability that leads to the variability in the WG strength and its density structure (Gordon et al., 2007;
Meredith et al., 2008; Gordon et al., 2010; McKee et al., 2011; Armitage et al., 2018; Gordon et al., 2020). However, the specific
thermodynamic mechanisms that link variability in surface forcings to AABW export remain unclear.

It is important to understand the drivers of variability in AABW formation and to quantify its variability. Its formation plays a
key role in the climate system as it provides a pathway for ventilation of abyssal waters, transport of nutrients and tracers (i.e.
nitrogen, phosphorous and oxygen), and storage for large amounts of carbon. Understanding how the deep ocean reacts to a
warming climate will give us insight to how oceans will contribute to sea level rise and CO2 content in the atmosphere. Purkey
and Johnson (2010) have shown that the 80 % radiative imbalance in the atmosphere that have gone into heating the ocean has




lead to a global change in abyssal heat content equivalent to adding 10 % to the total ocean heat storage, increasing rates of steric sea level rise by 8 %.

A crucial concept to understanding AABW formation variability, and ultimately the variability of the MOC, is water mass transformation (WMT). A water mass is defined as the water bounded by isosurfaces of a tracer. Any tracer can be used, but the most common tracers for WMT analysis are temperature, salinity, and potential density. The transformation of a water mass occurs when the water mass' density is altered through irreversible thermodynamic processes. In order for the MOC to exist, water masses must change density classes as they circulate between the surface and abyss, just like NADW and AABW. The

next section provides an overview of the WMT theory and explains its value in providing insight to global ocean circulation.

    While mooring observations such as described in Gordon et al. (2020) are useful for characterizing observed variability in water masses, they cannot provide a closed water mass budget, which requires dense observations of dynamic and thermodynamic flxues in space and time. So in order to investigate the dynamics and thermodynamics of temporal variability in AABW using the WMT framework, we turn to three state-of-the-art, data-assimilating ocean reanalysis products. Such products are a useful

tool in studying regions, such as the WG, that lack consistent and comprehensive observations, and for investigating physical mechanisms that drive variability. While reanalyses are far from perfect representations of the ocean state, they represent the best attempt to synthesize diverse observations in a consistent way. Even if physical processes such as coastal polynyas are not always represented accurately in ocean reanalyses, there is still value in understanding their internal dynamic and thermodynamic budgets. By diagnosing WMT in these reanalyses, we can probe the relationships between ocean surface fluxes with a changing

climate, how climate variability influences sea ice expansion and water mass transformation rates, and how that ultimately affects the abyssal water properties and circulation of the lower MOC cell. With datasets as described in section 3, and the water mass transformation (WMT) framework outlined by Walin (1982) (Sect 2), we strive to provide insight into the mechanisms and drivers of MOC variability in the WG.

    Our paper is organized as follows: In Section 2, the theory of WMT is introduced and we detail how the WMT budget was

calculated. In Section 3 we talk about the observational and model data used. Observational data from the World Ocean Atlas was used to compare to modelled bottom temperatures and salinities. The average, climatological and interannual variability of volume tendency, transport and transformation is discussed in Sections 4, 7, 5, respectively. Finally, the findings from the Section 5 is discussed and compared with similar studies in Section 6.

## 2   Water Mass Transformation Theory

Here we provide a brief introduction to the WMT framework, first employed by Walin (1982). The WMT-framework, in this context, allows for the separation of explicit mechanical and thermodynamic processes on ocean circulation (Walin (1982), Groeskamp et al. (2016), Groeskamp et al. (2019)) due to surface fluxes, advective transport, and diffusive mixing.

    A water mass is defined as the water bounded by isosurfaces of a tracer. Any tracer can be used, but the most common tracers for WMT analysis are temperature, salinity, and potential density. Here we use $\sigma_2$ - potential density referenced to 2000 dbar -

because of its ability to characterize stratification through the deep and abyssal ocean. (For notational simplicity, henceforth



we will write $\sigma$ instead of $\sigma_2$.) The limitation of investigating WMT through potential density is the inability to quantify the effect of cabbeling and thermobaricity on WMT, potential misrepresentation of neutral mixing (Iudicone et al., 2008b), and inability to distinguish water masses of the same density but different temperature / salinity (Evans et al., 2018). Here we define transformation of a water mass as the change in density of the fluid parcel due to its change in heat and salt content. Computing

the transformation budget for a basin or a remote region, such as the Weddell, can further our understanding by providing quantitative insight into the drivers of water mass variability.

  The potential density $\sigma$ of seawater evolves according to:

$$\frac{D\sigma}{Dt} = \dot{\sigma} = \frac{\partial \sigma}{\partial \theta}\dot{\theta} + \frac{\partial \sigma}{\partial S}\dot{S} \tag{1}$$

where the factors $\frac{\partial \sigma}{\partial \theta}$ and $\frac{\partial \sigma}{\partial S}$ are the thermal expansion and haline contraction coefficients, respectively. This expression includes

both a thermal component:

$$\frac{D\theta}{Dt} = \dot{\theta} = G^{\theta}_{hdiff} + G^{\theta}_{vdiff} + G^{\theta}_{surf} + G^{\theta}_{sw} \tag{2}$$

where $G^{\theta}_{hdiff}$ represents the temperature tendency due to lateral / isopycnal mixing, $G^{\theta}_{vdiff}$ is the tendency due to vertical/isopycnal mixing, $G^{\theta}_{surf}$ is the surface forcing, and $G^{\theta}_{sw}$ is the shortwave penetration; and a haline component

$$\frac{DS}{Dt} = \dot{S} = G^{S}_{hdiff} + G^{S}_{vdiff} + G^{S}_{surf} \tag{3}$$

where the terms represent the salinity tendency due to the same non-conservative terms as for temperature, except for shortwave radiation term. We use these forms of the heat and salt budget because they correspond to how those budgets are diagnosed from numerical simulations.

  To understand the equations to come, briefly we cover the Heaviside and delta functions. The Heaviside function, $\mathcal{H}$, is 0 when the argument is negative, and 1 for positive arguments. The Heaviside function is the integral of the delta function where

$$\frac{\partial \mathcal{H}(x)}{\partial x} = \delta(x) \tag{4}$$

This approach allows us to define a region under a reference isopycnal ($\tilde{\sigma}$). The density field is a function of $x, y, z$ and time, meaning

$$\sigma = \sigma(x, y, z, t). \tag{5}$$



so that the total volume of water *denser* than $\tilde{\sigma}$ within a region $R$ is given by

$$\mathcal{V}(\tilde{\sigma}, t) = \int_R \mathcal{H}(\tilde{\sigma} - \sigma) dV \tag{6}$$

i.e. the cumulative volume distribution.

The WMT budget for region $R$ in Eq. (7), expresses the relationship between the time evolution $\mathcal{V}$ to the inflow/outflow transports ($\Psi$) on the basin boundary, plus the thermodynamic transformation ($\Omega$) occurring in the basin interior:

$$\frac{\partial \mathcal{V}}{\partial t} = \Psi + \Omega, \tag{7}$$

where

$$\Psi(\tilde{\sigma}, t) = -\int_{\delta R} (\mathbf{u} \cdot \hat{\mathbf{n}}) \mathcal{H}(\tilde{\sigma} - \sigma) dA \tag{8}$$

($\delta R$ is the region boundary and $\hat{\mathbf{n}}$ is the unit normal on the boundary) and

$$\Omega(\tilde{\sigma}, t) = -\int_R \delta(\tilde{\sigma} - \sigma) \dot{\sigma} dV. \tag{9}$$

The $\dot{\sigma}$ in Eq. (9) comes from the potential density conservation equation (Eq. (1)). It represents the sum of the temperature and salinity components (Eqs. (2) and (3), respectively) of the non-conservative tendencies from horizontal and vertical diffusion, surface forcing and shortwave penetration (potential temperature only) (Abernathey et al. (2016), Supplemental Information).

The WMT framework helps us discern the contributions to the water mass variability by quantitatively relating it to the driving processes of surfacing forcing and advective transport to interior mixing.

### 2.1 Numerical Implementation

In order to assess the mechanisms behind AABW transformation and circulation variability, a fully closed WMT budget is desirable. To calculate this, we must first close the temperature and salinity budget. WMT volume budget analysis was conducted using ECCO, SOSE, and SODA reanalysis data. In all of these cases, the ocean is divided into discrete layers $\tilde{\sigma}_n$, and each of the relevant terms is calculated inside these discrete layers. The discrete analog of the cumulative volume integral (6) is

$$\mathcal{V}(\tilde{\sigma}_n, t) = \sum_{i,j,j} \Delta V \mathcal{H}(\tilde{\sigma}_n - \sigma) \tag{10}$$

where the summation is over each model grid cell within the region of interest and $\Delta V$ is the finite volume of each cell. Henceforth we use the shorthand $\mathcal{V} = \text{cumsum}_\sigma(\Delta V)$ to denote this operation.





The time-rate of volume change was computed and balanced by the calculated $\sigma$ total tendency ($\Omega_{tottend}$) and a residual (R1) due to discretization of isopycnal layers:

$$\frac{\partial \mathcal{V}}{\partial t} = \Omega_{tottend} + R1 \tag{11}$$

where

$$\Omega_{tottend} = -\frac{\partial}{\partial \tilde{\sigma}} \text{cumsum}_\sigma (G^\sigma_{tottend}) , \tag{12}$$

a where $G^\sigma_{tottend} = (\partial\sigma/\partial\theta)G^\theta_{tottend} + (\partial\sigma/\partial S)G^S_{tottend}$ is the model's total tendency for potential density, a weighted sum of total tendencies for temperature and salinity. This term is the sum of conservative tendencies due to horizontal and vertical advection, non-conservative tendencies due to horizontal and vertical diffusion, and surface forcings (including shortwave

penetration for the potential temperature component). It is helpful to calculate R1 explicitly in order to determine if the chosen $\sigma$ bin size is appropriate. We found that the budgets were not sensitive to changing $\sigma$ bin sizes. Smaller bin sizes meant higher resolution; however that did not significantly change R1 and increased computational costs.

Next, the total advection tendency term ($\Psi_{adv}$) is decomposed to $\sigma$'s velocity component ($\Psi_{vel}$) and a residual term (R2) attributed by a yet-to-be-determined combination of numerical mixing and numerical cabbeling effects:

$$\Psi_{adv} = \Psi_{vel} + R2 \tag{13}$$

The term

$$\Psi_{adv} = -\frac{\partial}{\partial \tilde{\sigma}} \text{cumsum}_\sigma (\frac{\partial\sigma}{\partial\theta} G^\theta_{adv} + \frac{\partial\sigma}{\partial S} G^S_{adv}) \tag{14}$$

represents the direct effect, as output from the models, of advection of temperature and salinity on the total tendency of potential density in each grid cell, while

$$\Psi_{vel} = -\text{cumsum}_\sigma (\mathbf{u} \cdot \hat{\mathbf{n}} \delta A). \tag{15}$$

is the net transport across the region boundary, accumulated in $\sigma_n$ bins; this is the conventional overturning streamfunction. In the continuous world, $\Psi_{adv}$ and $\Psi_{vel}$ can be shown to be mathematically identical. However, numerical discretization of the advection operator yields a residual term representing non-advective affects of the advection scheme. This residual has in fact been used to quantify numerical mixing in other studies (Holmes et al., 2021). Separating out the numerical residual allows us to

explicitly see the volume-transport of certain densities by purely physical inflow/outflow mechanisms, and by a mixing term.





Water mass transformation, $\Omega_{tf}$, was computed by summing the rest of the tendency terms due to non-conservative, thermodynamic processes such as diffusion, shortwave radiation and surface forcings of heat and freshwater:

$$\Omega_{tf} = -\frac{\partial}{\partial \tilde{\sigma}} \text{cumsum}_{\sigma} \left[ \frac{\partial \sigma}{\partial \theta} (G^{\theta}_{hdiff} + G^{\theta}_{vdiff} + G^{\theta}_{surf} + G^{\theta}_{sw}) + \frac{\partial \sigma}{\partial S} (G^{S}_{hdiff} + G^{S}_{vdiff} + G^{S}_{surf}) \right] \tag{16}$$

By defining residuals in this way, we arrive at numerical form of eq. 7 below which is closed by construction:

$$\frac{\partial \mathcal{V}}{\partial t} = \Psi_{vel} + R2 + \Omega_{tf} + R1. \tag{17}$$

Since R2 represents numerical mixing, we can more concisely write equation 17 to be

$$\frac{\partial \mathcal{V}}{\partial t} = \Psi_{vel} + \Omega^* + R1, \tag{18}$$

where

$$\Omega^* = \Omega_{tf} + R2. \tag{19}$$

SODA does not provide any tendency diagnostics; however, it does provide the velocity field. Thus in SODA we simply define $\Omega^*$ via the residual: $\Omega^* = \partial \mathcal{V}/\partial t - \Psi_{vel}$.

Equation 17) represents the WMT volume budget expressed by explicit terms due to physical and thermodynamic processes captured by the model. Now we can see the time-rate of volume change is balanced by the physical transport into and out-of the WG, plus the residual mixing term, the transformation term, and numerical discretization residual. Subsequent plots in sections 4, 4.2 and 5 display each term's contribution to the total WMT volume budget.

## 3 Data & Models

The observational data and numerical simulations used in this project are described here. The strengths and limitations of the models are mentioned in each section, and evaluation of each numerical model is discussed in section 3.5.

### 3.1 World Ocean Atlas

We used observational temperature and salinity as a baseline for validating the use of each model. Evaluation of each model compared with observations is in Sect. 3.5. The observational temperature and salinity data are from the World Ocean Atlas 2018 (WOA) (Zweng et al. (2018), Locarnini et al. (2019)). WOA is a set of 1° gridded climatological fields of in situ temperature and salinity. The 1° climatological fields averaged over the "climate normal" period 1981-2010, and the years 2005-2017 are presented in this paper as a means for model validation assessment.



Temperature and salinity profile data were obtained from bottled samples; ship-deployed Conductivity-Temperature-Depth (CTD); Mechanical, Digital and Expendable Bathythermographs (XBT), profiling floats, moored and drifting buoys, gliders, undulating oceanographic recorder (UOR), and pinniped mounted CTD sensors. WOA contains both observed levels profile data and standard depth profile data with various quality control flags applied. In most regions with sparse data coverage, such as the WG region, flagged data seen as outliers were not removed because they may still represent legitimate values; and are

therefore, included in the climatological periods used here (Zweng et al., 2018; Locarnini et al., 2019).

### 3.2   Estimating the Circulation and Climate of the Ocean (ECCO)

The Estimating the Circulation and Climate of the Ocean version 4 release 3 (ECCO) state estimate is a reconstruction of the 3-D time-varying ocean and sea ice state (Forget et al., 2015). Produced from MITgcm, ECCO has approximately 1° horizontal grid resolution and 50 vertical levels of varying thickness. It provides monthly-averaged 3-D ocean, sea-ice, and air-sea flux fields;

2-D daily-averaged ocean and sea-ice fields, and 6-hourly atmosphere fields, all covering the period Jan 1, 1992 to Dec 31, 2015.

    The ECCO state estimate provides a statistical best fit to observational data; however, unlike other ocean reanalyses that directly adjust the model state to fit the data, such as the Simple Ocean Data Assimilating model described in section 3.4, ECCO is a free-running model that simulates what is observed in the ocean based on the governing equations of motions, a set of initial conditions, parameters, and atmospheric boundary conditions. Some observational data ECCO uses are from Argo

floats, shipboard CTD and XBT measurements, marine mammals, mooring data, Radar Altimeter Database System, and NSIDC satellite products (Fukumori et al., 2018).

    The ECCO state estimate satisfies physical conservation laws, with no unidentified sources of heat and buoyancy. Due to the model's dynamically consistent nature, it conserves heat, salt, volume, and momentum (Wunsch and Heimbach (2007), Wunsch and Heimbach (2013)); and the state estimate can be used to explore the origins of ocean heat, salt, mass, sea-ice, and regional

sea level variability Forget et al. (2015). It uses a non-linear free surface combined with real freshwater flux forcing and the scaled height coordinate; and users of ECCO are able to assess model-data misfits (Forget et al., 2015). Known issues of the state estimate are mentioned for the first release in (Forget et al., 2015). For example, ECCO has residual systematic errors, especially in regions with sparse data (Buckley et al., 2017).

### 3.3   Southern Ocean State Estimate

The Southern Ocean State Estimate (SOSE) is a model-generated best fit for Southern Ocean observations (Mazloff et al., 2010). It is a solution to the MITgcm and a C-gridded dataset at 1/6° horizontal resolution, available at timestamps daily and annually (Mazloff, 2008). Its iteration runs at 5-day averages starting from Jan 1-5 2005, ending Dec 31 2010.

    SOSE is also a data-assimilating model. It uses similar observation data as ECCO. Some limitations are similar to that of reanalyses in regards of regions and variables being more reliable with more observations covered in those areas, and the

opposite being true for regions with sparse data. SOSE provides a self-consistent state estimate that satisfies momentum, volume, heat and freshwater conservation. Some of its key strengths are that SOSE has a better spatial and temporal resolution than most





state estimate models and is the highest-resolution model used in this study. It is dynamically consistent and best-fit to available
190 observations, and its biases are well-documented (Mazloff and National Center for Atmospheric Research Staff, 2021).

This study examines the Weddell Sea region, and so far one major bias has been found during our analysis: SOSE produces an
open-ocean polynya in the first year of its run (2005) which was not observed in the real ocean; therefore, this year is removed
from the time-mean and climatology, but the anomalous interannual variability still includes the year 2005.

### 3.4 Simple Ocean Data Assimilation

The Simple Ocean Data Assimilation ocean/sea ice reanalysis (SODA) is the third numerical model used for our water mass
transformation analysis. The SODA3 reanalyses are built on the Modular Ocean Model, version 5, ocean component of the
Geophysical Fluid Dynamics Laboratory CM2.5 coupled model (Delworth and Coauthors, 2012), with fully interactive sea ice on
a $0.25° \times 0.25°$ horizontal and 50-level vertical resolution (Carton et al., 2018). Improvements have been made in SODA3 such
as upgrades to the SST datasets and a 40 % increase in hydrographic data from the latest release of the World Ocean Database.
Earlier generations of ocean reanalysis have contained systematic errors that have several sources, including measurement bias,
inaccurate model physics and numerical resolution, and biases in fluxes and initial conditions. The release of SODA3 was an
effort to address these broad issues; for example, the adoption of the iterative flux correction procedure of (Carton et al., 2018)
addresses the bias in surface forcing in which flux error is estimated from the misfits obtained from an initial ocean reanalysis to
alter fluxes for a revised ocean reanalysis. SODA3 was also upgraded to be an ensemble reanalysis, for which the ensemble
spread provides an estimate of uncertainty. A comparison to ORAS5 and ECCOv4r3 is provided in Carton et al. (2019).

SODA3 is included with finer eddy-permitting spatial resolution, active sea ice, and bias adjustment. The version this study
uses is SODA3.4.2 (SODA henceforth), which means that the assimilated data is restricted to the basic hydrographic data and
SST with meteorological forcing derived from the European Centre for Medium-Range Weather Forecasts interim reanalysis
(ERA-Interim) daily average surface radiative and state variables (Dee, 2011) and the COARE4 bulk formula with flux bias
correction applied. The ocean and ice data runs every 5 days running through Jan 4, 1993 to Dec 19, 2019, and the transport files
every 10 days from Jan 7, 1993 to Dec 17, 2019. There were jumps in the salt field from the ocean files occurring before 1997
that we suspect is the result of the reanalysis's nudging technique. For this reason, we have used SODA running from February
15, 1997 to Dec 17, 2019.

The biggest difference between SODA and ECCO and SOSE is their method of data assimilation. ECCO and SOSE use the
adjoint data assimilation method which optimizes the initial conditions and model parameters by incorporating observation data
to a physics-based numerical simulation. The SODA experiment uses an optimal interpolation method for their data assimilation
in which the ocean state is constructed from a forecast using a linear deterministic sequential filter and based on the difference
between observations and the forecast mapped onto the observation variable and its location (Carton et al., 2018). Due to the
method of data assimilation, it is not possible to diagnose a closed heat budget in SODA, and therefore we cannot explicitly
calculate $\Omega$. However, we can calculate $\partial V/\partial t$ and $\Psi$. So in the following analysis, we infer $\Omega$ as a residual.





## 3.5 Model Assessment

It is well known that numerical models struggle to capture the complex physics on the Antarctic shelves that determine AABW properties and circulation (Heuzé et al., 2013). Although these models assimilate data, they are still constrained by their resolution and physical parameterizations. In this section we assess each model against the World Ocean Atlas data. This step gives a base for assessing each model's reliability in simulating ocean physics in a region with limited observational data, such as the Weddell Sea region. Such validation was assessed by comparing bottom temperature and salinity spatial distributions,

as well as time-averaged Temperature Salinity (TS) distributions, in the WG region between ECCO/SOSE/SODA and WOA time periods 1981-2010 and 2005-2010. Only the climate normal period (1981-2010) is presented here since the spatial and TS variability of both WOA time periods are practically identical in this region.

Figure 1, (a), (c), (f) and (i) shows the time-mean bottom temperatures in the WG region for WOA (1981-2010), ECCO (1992-2015), SOSE (2006-2010) and SODA (1993-2019), respectively. The coldest temperatures in each model are located near

the coast of Antarctica and gradually becomes warmer equator-ward. However, in ECCO there appears to be a strip of relatively warmer water enclosing the coldest temperatures before the more gradual equator-ward warming is displayed. From Fig. 1 (d), (g) and (j), which shows the standard deviation of temperature from ECCO, SOSE and SODA, respectively, we can see that in ECCO and SODA (Fig. 1 (d) and (j)) there is little spatial variability in bottom temperature; whereas in SOSE (Fig. 1 (g)), there is a lot of variability in the open region of the Weddell. Furthermore, looking at the difference in temperature between

each model and WOA (Fig. 1 (e), (h) and (k)), we also see that ECCO and SODA are overall warmer than observations (Fig. 1 (e) and (k)). From ECCO and SODA, the difference is as high as 2° C; however, in SODA, the model on average simulates colder bottom temperatures along the Antarctic Peninsula (Fig. 1 (k)). In SOSE, however, there is less of a difference between simulated and observed temperature (Fig. 1 (h). The region with hash marks denotes the places where the difference between modelled and observed temperatures is less than the standard error of the observational estimates; and as we can see in SOSE

(Fig. 1 (h)), the model's bottom temperature agrees with WOA in the open ocean.



**Figure 1.** Annual mean of bottom temperatures of the Weddell Gyre region for (a) WOA (1981-2010), (c) ECCO (1992-2015), (f) SOSE (2005-2010) and (i) SODA (1993-2019). Standard error of WOA (1981-2010) observations is shown in (b); and standard deviations are shown for (d) ECCO, (g) SOSE and (j) SODA. The differences between the observed and simulated bottom temperatures are shown in (e) ECCO, (h) SOSE and (k) SODA. The black hash lines denote the areas where the difference between model and WOA is less than the observation's standard error.





Following a similar assessment for bottom salinity, the salinity distribution in all three models and WOA appears to be more uniform than their respective temperature fields. The distribution in SOSE, Fig. 2 (f), indicates that the model is on average fresher than the others. Another distinction that stands out is the more variable spatial distribution in ECCO and SODA's salinity field (Fig. 2 (c) and (i), respectively), with fresher water located closest to the coast of the continent. In ECCO, however, there

are saltier plumes under the Ronne-Filchner ice shelf region (Fig. 2 (c)) indicating that ECCO is reproducing High Salinity Shelf Water (HSSW) in around the same areas present in the observational data (Fig. 2 (a)). Again in ECCO and SODA, we observe little spatial variability overall in bottom salinity (Fig. 2 (d) and (j)); however, in SOSE (Fig. 2 (g)) and SODA, there are strong deviations from the average salinity field near the tip of the Peninsula. In SODA, there is also significant variability in the eastern part of the region along the coast. In Fig. 2 (e) and (k), there is very little difference between model and observational

bottom salinity, as is indicated by the black hash marks. The biggest difference we see between model and observed salinity is between SOSE and WOA. As we saw from Fig. 2 (f), SOSE is much fresher than what is observed and simulated in the other models, by about 0.5 psu.



**Figure 2.** Annual mean of bottom salinity of the Weddell Gyre region for (a) WOA (1981-2010), (c) ECCO (1992-2015), (f) SOSE (2005-2010) and (i) SODA (1993-2019). Standard error of WOA (1981-2010) observations is shown in (b); and standard deviations are shown for (d) ECCO, (g) SOSE and (j) SODA. The differences between the observed and simulated bottom salinities are shown in (e) ECCO, (h) SOSE and (k) SODA. The black hash lines denote the areas where the difference between model and WOA is less than the observation's standard error.





To further test the validity of model representation of real world processes, volume-weighted temperature-salinity (TS) distributions were compared between model and observations. Once again, only the climate normal period is presented in the comparison since both time periods used were nearly identical. Fig. 3 displays the volume-weighted, TS diagrams of WOA (1981-2010), ECCO state estimate, SOSE and SODA. The difference between each model's TS distribution and observation is shown in Fig. 4. The approximate temperature and salinity profile ranges of AABW are as follows: -0.2° C < $\theta$ < -0.7° C and 34.6 psu < S < 34.7 psu (Carmack, 1974; Mackensen et al., 1996).

Looking at Fig. 3, an obvious observation is that all three models' TS distribution have similar shape, with SODA having more spread over the lighter density ranges. WOA's spread across density contour 1037.0 kg m$^{-3}$ and lighter, however, is unlike any of the models' TS distribution. Observation and models display that the most voluminous waters in the WG are CDW and AABW since their densities and temperature/salinity ranges exist in those regions (darkest blue).

We took the difference between each model's and observed TS distribution and viewed the difference on a semi-logarithmic scale, Fig. 4. This allows us to see a wider range of data. All models under-represent the coldest, densest HSSW in TS space. This suggests biases in the AABW formation processes.





**Figure 3.** Averaged temperature-salinity distribution of the WG region for WOA (1981-2010), ECCO, SOSE and SODA.




## TS Diagrams
## Difference between model and WOA (1981-2010)

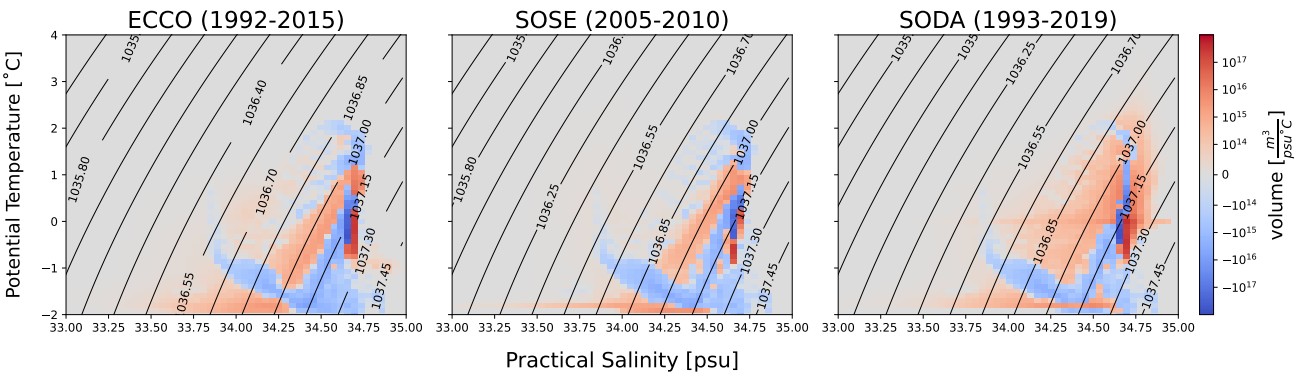

**Figure 4.** Averaged temperature-salinity distribution of the difference between WOA and each model.

Overall, ECCO and SODA appear to reproduce bottom salinity where the difference between modeled and observed values is less than the observation's standard error (as denoted by the hash marks). SOSE reproduces bottom temperature with the difference less than WOA's standard error as well. Simulated bottom temperature in ECCO and SODA, however, is warmer than what is observed; and SOSE has a fresher tendency than what is observed. These are important biases in each model to note as we continue our analysis of AABW variability using all three models.

## 4 Climatological WMT Budgets

### 4.1 Annual Mean

We first examine the annual-mean WMT budget, which represents the long-term mean transformation and overturning structure of the WG, averaging over all spatial and temporal variability. Figure 5(a), (b) and (c) shows the time mean budget from ECCO, SOSE and SODA, respectively. The black line represents the time evolution of the cumulative water mass volume distribution, $\frac{\partial V}{\partial t}$, in the WG region during each model's respective time period. It is balanced by the total inflow/outflow transports ($\Psi$, red line) and mean transformation ($\Omega^*$, green line), which includes the residual due to numerical mixing. Conceptual models usually assume that the system is in a steady state balance between overturning and transformation ($\partial V/\partial t = 0$), but that is clearly not the case for any of the models examined here. The existence of a tendency indicates a model drift and / or low-frequency variability over the climatological period. The grey dashed line, R1, represents the residual term due to discretizing the volume distribution. Figure 6 (a) and (b), ECCO and SOSE, respectively, shows the transformation term ($\Omega^*$) decomposed into mixing (purple dashed line) and surface components. The residual due to numerical mixing, R2, is shown by the pink dashed line in Fig. 6. The surface-induced transformation is further broken down into temperature (blue dashed line) and salinity (orange dashed line) components; which can be interpreted as cooling/warming-induced transformation and freshwater flux-induced



transformation, respectively. The reader should note that the $\Omega^*$ transformation term in SODA lumps together all the residuals that were explicitly calculated in ECCO and SOSE (R1 and R2).

In $\sigma_2$ coordinates the water masses that are the Weddell's recipe for AABW are distinguished by their densities as follows: CDW/WSDW ($1037.13 < \sigma_2 < 1037.24$ kg m$^{-3}$) and WSBW/HSSW/ISW ($\sigma_2 \geq 1037.2$ kg m$^{-3}$). Here, the generic term of AABW is chosen to embody the total volume of Weddell bottom waters that are denser than the CDW of the Antarctic
Circumpolar Current. The denser precursory constituents of AABW (i.e. High Salinity Shelf Water (HSSW), Ice Shelf Water (ISW), CDW/Weddell Sea Deep Water (WSDW) and Weddell Sea Bottom Water (WSBW) are all contained within the range $\sigma > 1037.0$ kg m$^{-3}$. On average, as shown by the negative extremum of $\Psi$, AABW is exported from the region at the rate of 7.6 Sv in ECCO, 11 Sv in SOSE, and 2.7 Sv in SODA. The export is largely balanced by thermodynamic transformation ($\Omega$) by 6.1 Sv in ECCO, 0.9 Sv in SOSE, and 4.2 Sv in SODA. This leads to a total volume loss of the bottom water class of 1.34Sv in
ECCO, 7.85Sv in SOSE, but a total net volume gain of 1.53Sv in SODA.

Export values from ECCO and SOSE agree with the mean outflow value of Kerr et al. (2012) who found mean outflow of AABW using a 1/12° 20-yr global ocean simulation to be $10.6 \pm 3.1$Sv. Similar transport value of AABW have been reported by Talley et al. (2003), who determined 8.5Sv of AABW traveling northward in the Atlantic sector of the Southern Ocean meridional overturning circulation.

We now examine the thermodynamic processes driving transformation in more detail. AABW transformation in ECCO (Fig. 6 (a)) is mainly due to brine rejection at the surface (orange dashed line). The effects of surface cooling and mixing are generally confined to lighter water masses in the region and tend to cancel each other out. However, mixing does have a stronger tendency to lighten bottom water (by about 2Sv) than surface cooling induced positive transformation (1 Sv). In ECCO, the dominant impact of brine rejection on transformation (5.72Sv) is consistent with Iudicone et al. (2008a) who found brine
rejection dominating transformation of bottom water by ∼5Sv. The volume gain of bottom water in SOSE (Fig. 6 (b)) due transformation is weaker than in ECCO and is driven by an almost equal combination of surface cooling (0.86sv) and brine rejection (0.54Sv), countered by mixing (0.47Sv). Overall, the ECCO and SOSE models are in agreement with the literature of what is known about bottom water circulation in the Weddell Sea (Carmack and Foster (1975), Orsi et al. (1999), Meredith et al. (2000), Gordon et al. (2001), Naveira Garabato et al. (2002)). The two models show that brine rejection and surface cooling
drive positive bottom water transformation while mixing with warmer WDW and fresher AAIW acts to lighten the the water mass' density. Because we cannot explicitly calculate WMT in SODA but rather infer it as a residual, it is not possible to further decompose SODA's transformation into different components.





**Figure 5.** Main terms of the WMT-volume budget in $\sigma$ space averaged over each model's respective time period for **(a)** ECCO (1992-2016), **(b)** SOSE (2006-2010) and **(c)** SODA (1993-2019). Volume change is represented by the black line, transport by the red line, transformation is the green line (includes numerical mixing in ECCO and SOSE, and in SODA also includes discretization residual), and the discretization residual is explicitly shown as the grey dashed line in ECCO and SOSE.




**Figure 6. (a)** Transformation term, $\Omega$ (green solid line), broken down into its sources of transformation: surface salt flux (orange dashed line); surface heat flux (blue dashed line); and mixing (purple dashed line). Residual due to numerical mixing is the pink dashed line in ECCO (a) and SOSE (b).

## 4.2 Seasonal Climatology

While the long-term annual mean shown above is what matters most for the global-scale overturning and climate, the annual
mean masks a huge amount of seasonality in WMT. Exploring this seasonality is important for understanding the mechanisms behind WMT. Here we examine the monthly climatology of the transformation budget in the three models. The climatology of each term was calculated as a monthly average. Fig. 9 shows this climatology via contour plot, introducing an additional "month"



dimension to the WMT budget; the mean over all months corresponds exactly to the annual climatologies of the previous section. For ECCO and SOSE, we also decomposed $\Omega$ into transformation from mixing, surface cooling/warming and surface freshwater
flux.

Overall, the monthly view reveals that the overturning $\Psi$ is relatively steady over the year in all three models. However, both $\Omega$ and $\partial V/\partial t$ exhibit a seasonal cycle an order of magnitude larger than their annual mean. Moreover, the seasonal cycle in these terms is largely compensatory; excess dense water is created by WMT in winter and then destroyed in summer. Only a relatively small residual is left over for to export from the basin via $\Psi$. Breaking down the components of $\Omega$ into surface salt and
heat fluxes interior mixing, we can see some significant difference between ECCO and SOSE. In ECCO, the contribution of $\Omega^\theta_{surf}$ is negligible in the AABW range, and $\Omega^S_{surf}$ is positive throughout the winter; this indicates that brine rejection is the dominant process behind WMT, with little impact from ice melt, runoff, or precipitation. SOSE, in contrast, contains some negative values of $\Omega^S_{surf}$, corresponding to surface freshening, as well as a large surface cooling effect on WMT, in the AABW range. It is interesting to note how both models exhibit broadly similar $\sigma$-temporal structure in these terms, but with different
magnitude and position within $\sigma$ space.

Before moving on to look at interannual variability, we wish to focus on a single timeseries that best represents AABW transformation and overturning, rather than the entire range of densities. We do this by moving from a *transformation budget* to a *formation budget*. Specifically, for each model, we define the boundary between CDW (inflowing water) and AABW (outflowing water) as the density $\sigma^\dagger$ where $\Psi$ reaches is extreme (minimum value). The values of $\sigma^\dagger$ are 1037.125 kg m$^{-3}$ for ECCO and
SOSE and 1037.195 kg m$^{-3}$ for SODA. By sampling $\Psi$, $\Omega$, and $\partial V/\partial t$ at $\sigma^\dagger$, we obtain a single value or timeseries representing the net export rate, formation rate, and volume tendency of AABW.

These monthly climatology timeseries, shown in Fig. 7, largely reveal a similar picture to those seen in Fig. 9, in a more compact way. Again, in all three models in Fig. 7, we can see from the black line, $\partial V/\partial t$, that bottom water is gaining volume during the austral winter months and losing volume during the rest of the year. However, in SODA (Fig. 7c), bottom water
volume gain occurs over a longer time range, for three quarters of the year. Looking at $\partial V/\partial t$ over a wider density range in Fig. 9, we can see in all three models that densification acts upon the lighter densities in the beginning of winter, and as the season progresses the thermodynamic process moves down to transform the denser range of the ocean. This pattern can be derived from the surface cooling component of transformation. This is in agreement with the general understanding that AABW in the Weddell is created when sea ice forms during the winter season, and during the warmer months the production of AABW
decreases (Fig. 7 (a)) or is even being destroyed (Fig. 7 (b) and (c)). There is year-round export of bottom water in all three models, with the intensity of outflow peaking and varying in the middle of the year. Peak export happens at ~10Sv in June in ECCO, ~12Sv April-July in SOSE, and ~8Sv May-July in SODA. Both ECCO and SOSE values are within the error bars of Kerr et al. (2012)'s maximum monthly mean outflow value of $12.2 \pm 3.0$Sv. SOSE reveals an overall imbalance between the formation rate of AABW and its export; more water is being exported than is being formed. This leads to a continuous decrease
in AABW volume (also visible in Fig. 5 (b). However, superimposed upon this overall trend is a strong seasonal cycle, with excess AABW production and corresponding volume tendency in winter.





In ECCO, as we saw in the time-mean budget, Fig. 8 (a) and Fig. 9 (a) also shows that bottom water volume gain is mainly due to brine rejection. Surface cooling plays a tertiary role as it acts just during the second half of the winter season. All the while, throughout the year, transformation due to mixing is trying to homogenize the waters; therefore, bottom water is essentially being destroyed throughout the year - on the order of 4 Sv. Sea ice formation is most intense from May to August and encourages bottom water formation especially within the 1037.16 - 1037.20 kg m$^{-3}$ density range. This is concurrently being cancelled by the constant mixing which ultimately leads to less bottom water transformation, as can be seen in the summary of these dynamics in $\partial V/\partial t$.

In SOSE, AABW production during the winter months is also mainly due to brine rejection. Surface cooling provides an additional source for AABW formation. Mixing, though not as significant, during late winter month adds to AABW volume as well.

One immediately notices in SODA the overall larger monthly variability, and the dramatic switch between volume gain in January to significant volume loss in February (>25 Sv). SODA, as it was detailed in Sect. 3, differs from the other two data assimilating models in that SODA nudges the data to fit the observations it is trying to match. Such data assimilating technique commonly causes jumps in the data which is consequently evident in our water mass transformation budgets. For this reason, SODA's time dimension was cropped to exclude the years prior to 1997 where significant jumps in salinity data were avoided. It would help to remind the reader that the $\Omega^*$ term shown here is not quite the same as the transformation terms shown for ECCO and SOSE. The transformation term in SODA implicitly carries the residuals that were explicitly calculated in ECCO and SOSE (R1 and R2). In SODA, the advective transport of bottom water is almost insignificant compared to the transformation that arises from the budget. SODA still concurs with ECCO and SOSE, showing that volume gain occurs during the winter months; however, bottom water volume grows over half a year as opposed to only a few months centered in the year in ECCO and SOSE.





**Figure 7.** Monthly climatology of Weddell bottom water's main WMT budget terms in ECCO (a), SOSE (b) and SODA (c). Residual due to discretization (R1) only shown in ECCO and SOSE.





**Figure 8.** Monthly climatology of Weddell bottom water's transformation term (green line), its sources of transformation: surface salt flux (orange dashed line); surface heat flux (blue dashed line); and mixing (purple dashed line). Residual due to numerical mixing is the pink dashed line in ECCO (a) and SOSE (b).






**Figure 9.** Monthly climatology of main WMT budget terms in top panel in ECCO (a), SOSE (b) and SODA ((c); and sources of transformation (bottom panel) in ECCO (a) and SOSE (b). The horizontal black line represents the bottom water boundary in each model (1037.125 kg m$^{-3}$ for ECCO and SOSE, and 1037.195 kg m$^{-3}$ for SODA).



## 5 Interannual Variability

Having examined the climatology, we now turn to the main focus of our study: quantifying the interannual variability of AABW production in these reanalyses. A deep look at the interannual variability of the anomalous WMT budget terms reveals some interesting differences of AABW circulation between these three models and also serves as a point of comparison with similar studies (Iudicone et al., 2008a, b; Abernathey et al., 2016; Gordon et al., 2020). In this section, we highlight some anomalous events and the differences between all three models.

The anomaly timeseries for each model are constructed by sampling $\Psi$, $\Omega$, and $\partial V/\partial t$ at $\sigma^\dagger$, then removing the monthly climatology for each term, and smoothed by a rolling mean with the center window set to the middle of the year; these are shown in Fig. 10. For ECCO and SOSE, we also decompose $\Omega$ into components due to surface heating, surface salt flux, and interior mixing in Fig. 11. Figures 10 and 11 are thus completely analogous to Figs. 7 and 8, but for the interannual variability, rather than the seasonal cycle. The anomaly WMT timeseries each cover a different time period. SODA is the longest and provides the most recent data; the data we extracted begin in 1993 and go through 2020. ECCO begins in 1992 and extends through 2016. SOSE's time period is the shortest out of the three models; while it is the highest-resolution product analyzed here, its six-year period offers only a very limited view into interannual variability.

The most immediate feature that jumps out from Fig. 10 is the fact that the magnitude of the variability of $\partial V/\partial t$ in SODA is 10x larger than ECCO and SOSE, with values as large as 80 Sv. This indicates that the volume of AABW in SODA is changing by huge amounts from year to year. These changes are not explainable by variations in overturning $\Psi$ so they must be balanced by the inferred WMT $\Omega^*$ (recall that we cannot diagnose $\Omega$ directly from SODA but instead infer it as a residual). We hypothesize that these large-magnitude changes in AABW volume are due to SODA's data assimilation nudging the temperature and salinity fields, without a driving physical process.

The ECCO timeseries (Fig. 10(a)) reveal a moderate degree of variability (approx. 2-4 Sv) in the WMT budget and a varying relationship between terms. In some years (e.g. 1998) an increase in AABW volume is driven only by increased formation, likely due to mixing with watermasses precursory to AABW. While in late 2002, the increase in AABW is driven by increased formation and inflow. In others, (e.g. 2003), a decrease in AABW is driven by an increase in outflowing water. Whereas, in late 2010-2012, the decrease in AABW volume was caused by a strong negative transformation. The discretization residual term $R_1$, negligible in the climatological budgets, is generally also small in the ECCO timeseries, but does appear significant between 2004-2012, a fact for which we have no explanation.

Digging into the decomposition of WMT terms in ECCO (Fig. 11(a)), we see, somewhat surprisingly, that WMT variability is largely driven by variability in mixing, rather than air-sea-ice interactions. The one exception is an event occurring from 2004-2008 associated with anomalously strong surface cooling. (Surface heat fluxes are otherwise negligible in the AABW budget for ECCO.) Our initial assumption during this 2004-2008 period was that a polynya occurred - as is the case for SOSE in 2005. However, surface heat flux and sea ice cover maps with overlaid $\sigma_2^\dagger$ contours (not shown here) show no evidence of a simulated polynya, but do show outcropping of AABW contour in the open region of the Weddell. We suspect that internal





dynamics in ECCO drive AABW to outcrop under sea ice, where it loses heat. The heat loss due to this interaction is largely compensated by heat gain due to mixing, likely associated with convective mixing in the outcrop region.

SOSE is unique among the three models in showing a persistent trend in the WMT budget over its (relatively short) time period. As shown earlier in Fig. 2(b), SOSE is losing AABW and gaining CDW over this time period at a rate of 8 Sv, comparable to its overturning rate of 10 Sv. In the anomaly timeseries (Fig. 10(b)), we see that this trend is not steady but in fact accelerates

over time. Production of AABW ($\Omega$) is anomalously strong in the beginning of the state estimate (2005) and decreases by 14 Sv by 2010. Fig. 11 shows that this decrease in production rate is driven largely by trends in mixing. In the first year of the state estimate, an open-ocean polynya events contributes to enhancing production of AABW via surface heat fluxes; after that, there is no anomalous surface-heat-flux-driven production. There is some variability in production from surface salt fluxes of order 4 Sv. During the same time period, export of AABW ($\Psi$) decreases by 7 Sv (note that Fig. 10 shows the anomaly

relative the climatology, not the absolute value). These two trends partly cancel each other, but their residual imbalance drives an acceleration in the rate of volume loss of AABW. Overall, SOSE is the farthest of all the models from a state of balance between production and export AABW; production of AABW is clearly unable to keep up with export, and this gets worse, not better, over the estimation period.





**Figure 10.** Anomalous main terms of WMT-volume budget in ECCO (a), SOSE (b) and SODA (c). Timeseries was smoothed by an annual rolling mean with the center window set to the middle of the year. Volume change is the black line, transport the red line, transformation is the green line (includes R2 in ECCO and SOSE, and in SODA also includes R1 and R2), and the discretization residual is explicitly shown as the grey dashed line in ECCO and SOSE.



# Anomalous transformation

**Figure 11.** Anomalous transformation term, $\Omega$ (green solid line), broken down into its sources of transformation: surface salt flux (orange dashed line); surface heat flux (blue dashed line); and mixing (purple dashed line). R2 is the pink dashed line.

In general, the timeseries in Figs. 10 contain many different relationships between the different budget terms in different
models. To try to summarize these relationships, we compute correlations between each pair of terms. These correlations
quantify the extent to which one term balances the other in a budget (Tesdal and Abernathey, 2021); for example, a correlation of
1 between $\partial V/\partial t$ and $\Omega$ means that trends in volume distribute are completely driven by anomalous water mass transformation.
For ECCO (green bars), $\partial V/\partial t$ correlated strongly with transport $\Psi$ (0.66) and also with transformation (0.55). The correlation




## Correlation values for each model's anomalous budget terms

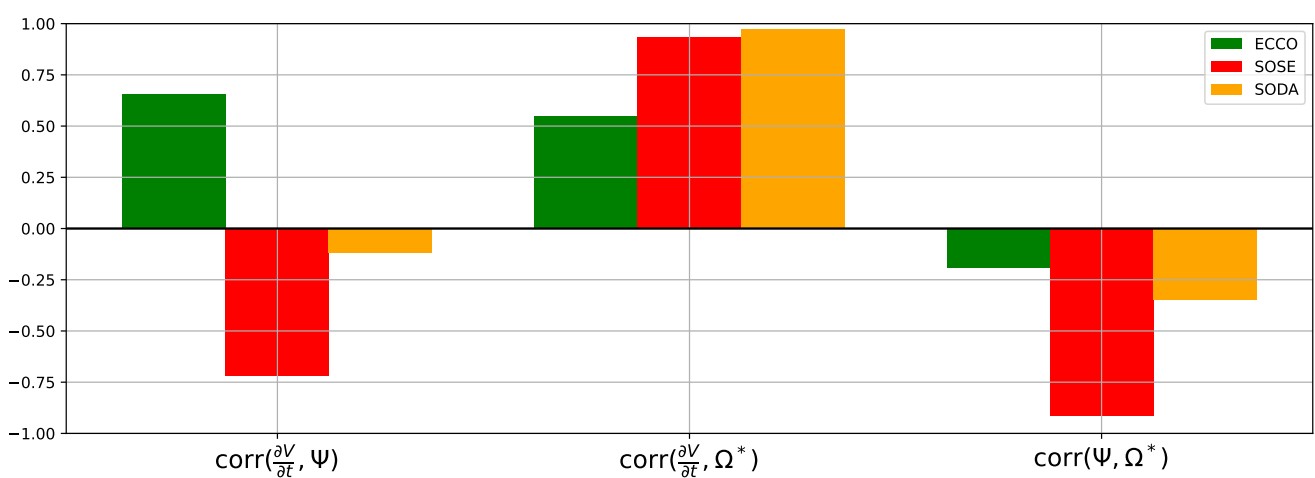

**Figure 12.** Correlation between the main budget terms from each model. The bars in green represent the correlation values in ECCO, red in SOSE and orange in SODA.

between $\Psi$ and $\Omega$ is weakly negative (-0.19), indicating moderate compensation between these two terms. In SOSE (red bars),
there is a very strong negative correlation between transformation and and transport (-0.91). Unlike in ECCO, there is a stronger
relationship between trends in volume and transformation (0.94) relative to transport (-0.72); in SOSE, volume trends in AABW
are driven by variability in both transformation AND transport. Finally, in SODA (orange bars), variability is completely driven
by transformation (0.97), and has no relationship to transport (-0.12). There is a weak anti-correlation between transport and
transformation (-0.35).

Overall, the correlation analysis confirms what is seen visually in the timeseries; each reanalysis has a very different overall
relationship between WMT budget terms.

Many studies have argued for links between large-scale climate indices and AABW production / export (Gordon et al., 2007;
Meredith et al., 2008; Gordon et al., 2010; McKee et al., 2011; Armitage et al., 2018; Gordon et al., 2020). We examined this
in the three reanalyses by calculating correlations between the terms in the WMT budgets and climate forcings from El Niño
Southern Oscillation (ENSO), the Southern Annular Mode (SAM), wind stress curl (WSC) and sea ice concentration (SI). ENSO
data was taken from the NOAA Extended Reconstructed Sea Surface Temperature (ERSST) version 5 project (Huang et al.,
2017). The SAM index was obtained from Marshall and National Center for Atmospheric Research Staff (2021). Windstress
curl was calculated from ERA-Interim zonal wind stress state variables (Dee, 2011), and finally, for sea ice concentration we
used each model's sea ice concentration diagnostic averaged over the WG region. We standardized each index by dividing the
anomaly timeseries by their respective standard deviation in time. Notable correlation values in SOSE are between transport and
SI (0.67) and WSC (-0.33). Transformation was negatively correlated with SI (-0.56). In SODA, transport is positively correlated
with SI (0.45). In ECCO, there were no notable correlation between budget terms and climate indices.



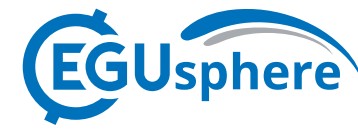

## 6 Discussion & Conclusion

Overall, the main contribution of our study is to diagnose, for the first time, a closed, time-dependent water mass budget for
AABW in the Weddell Sea from three state-of-the-art ocean reanalyses. This gives an unprecedented view into the processes
that control the volume, production, and export of AABW from this climatically important region. In the long-term annual mean,
all three models produce and export AABW at rates broadly compatible with observations. However, there is little agreement
between them at the level of anomaly timeseries.

We now summarize some of the main features of the AABW volume budget timeseries in each reanalysis. SODA showed an
extreme amount of variability in AABW volume in the WG which could not be explained by variations in export. Although we
could not diagnose WMT explicitly in SODA due to its lack of closed heat and salt budgets, the only possible explanation for
this variability is WMT, likely driven by the nudging tendencies of the data assimilation scheme. The obvious conclusions is
that reanalyses based on 3DVar data assimilation are not suitable for WMT studies, since they are not constrained to actually
conserve heat and salt. Due to their adjoint-based data assimilation, ECCO and SOSE do provide such closed budgets, and
therefore can provide better insight into the drivers of variability in WMT. Both models show strong interannual variability in
the AABW volume budget. SOSE's short time period makes it hard to draw general conclusions; during this time period, there is
an accelerating loss of AABW, driven largely by interior mixing and changes in surface salt fluxes. The transformation changes
are partly, but not completely, offset by a decline in export. Given the short time period of SOSE, these trends may simply be a
part of low frequency interannual variability. Indeed ECCO does display such interannual variability; there are numerous 5-year
periods that show secular trends in one or more terms. Moreover, there is some indication of alignment in these trends between
ECCO and SOSE, particularly the strong decline in AABW production from 2007 - 2010. In our assessment, ECCO provides the
most useful timeseries for revealing the processes and mechanisms that drive WMT and export variability. It exhibits interannual
fluctuations about its mean state with a reasonable magnitude relative to the climatology. The decomposition of WMT in ECCO
reveals a rich interplay between variability in export, surface forcing, and interior mixing in driving AABW volume variability.

Because of the difficulty of observing the deep outflow of AABW, it would be very useful if we could relate AABW export
to surface processes in the Weddell Sea. Gordon et al. (2020) showed strong interannual variability in Weddell Sea Bottom
Water salinity, as measured by nearly 20 years of mooring data in the northwest Weddell basin. They made the case that this
variability was tied to the strength of the WG and ultimately the wind stress curl, which is influenced by large-scale climate
modes such as ENSO and SAM. In a similar vein, Kerr et al. (2012) found a strong co-varying relationship between bottom
water transport and brine rejection in a 20-year high-resolution numerical simulation. We searched for such relationships in
ECCO and SOSE. We found transport in SOSE to be anti-correlated with: surface salt flux-induced transformation (-0.83);
surface heat flux-induced transformation (-0.62); and mixing-induced transformation (-0.91). Both this study and Kerr et al.
(2012) find that salt fluxes near the surface and bottom water export co-vary as they are both directly influenced by wind forcings
that influence WG strength. We found the windstress curl over the WG region to be most correlated with the surface salt flux
source of transformation (0.47) in SOSE. However, SOSE is very short, so we must take these correlations with a grain of salt.
In contrast, there was no strong relationship between transport and the different sources of transformation in ECCO. We note





the importance of the volume tendency term $\partial V/\partial t$ in the WMT budget. When this term is large, it means that excess AABW production need not correspond with export; instead, it can drive large trends in the water mass distribution within the basin, without any impact on export. Over its 25-year timeseries, the relationship between the terms in the WMT budget changes

dynamically, with no clear dominant balance. This complexity confounds the goal of establishing a simple relationship between surface forcing and AABW export. This is an important insight for studies of the WG based on surface and satellite-based observations.

Numerous studies have shown that AABW in the Weddell Sea has been warming (Robertson et al. (2002), Fahrbach et al. (2004), Purkey and Johnson (2010), Fahrbach et al. (2011)), freshening (Jullion et al. (2013)), and losing volume (Purkey and

Johnson, 2012; Kerr et al., 2018). Purkey and Johnson (2012) suggest that the decline in AABW production by a rate of -8.2 (±2.6Sv) is the cause for the contraction of bottom water volume for the period 1993-2006. As shown in the annual mean (Fig. 2), both ECCO and SOSE are losing AABW volume, with SOSE losing AABW much faster. SODA, in contrast, shows an increase in bottom water volume. Because of how SODA's data assimilation works, it is impossible to trace this trend back to specific physical processes. In both ECCO and SOSE, the decline in AABW is due to an imbalance between AABW formation

and export; both models export more AABW than they produce. In ECCO, the volume loss is roughly steady over 25 years, with significant interannual variability. In SOSE, in contrast, it accelerates strongly over the 6-year period.

In Purkey and Johnson (2012), it is hypothesized that the freshening of the northwest shelf water contributing to an increase in glacial meltwater input is the cause of the slowdown of AABW production rate (Hellmer et al., 2011). Surface freshening makes it more difficult for surface and shelf waters to sink which slows the production of AABW and subsequently, the circulation of

the lower limb of the MOC. In SOSE, Fig. 11 (b), there is a declining trend in transformation, partly due to a switch from brine rejection inducing formation prior to mid-2008, to freshwater input causing AABW destruction from mid-2008 to mid-2009. However, the surface salt flux source of transformation switches back to brine-rejection, inducing formation of AABW. Since SOSE does not include time-variable runoff, such trends must be due to variations in sea ice or precipitation. In ECCO, we do not see such a trend in transformation nor in freshwater flux-sourced transformation to suggest a decline in AABW volume due

to freshening.

Our project has sought to fill our gaps in understanding of the thermodynamic drivers of AABW export variability from the Weddell Sea; specifically, the quantitative links between surface forcing, interior dynamic and thermodynamic processes, and outflow. From the WMT analysis employed with the three ocean state estimates, we have determined that the variability of AABW is driven by a combination of surface forcing derived from strong winds and brine rejection and interior diapycnal

mixing. An additional initial goal of our work was to probe the mechanistic link between climate forcings, SAM and ENSO, and AABW transformation and export, as has been suggested by observations (Gordon et al., 2020). However, none of the reanalyses we analyzed exhibited such clear links. We further had hoped that, since these reanalyses assimilate data and aim to capture the real history of the ocean state, they might simulate the specific phasing of interannual variability seen in the mooring records; however, this was not the case. The discrepancies between the models, and between the models and observations, suggest that

this class of reanalyses is not capable of consistently and faithfully capturing the processes that drive AABW variability in a robust way. Similar to the conclusion of Heuzé et al. (2013) regarding CMIP5 models, the relatively coarse resolution of these



models makes it difficult to resolve processes such as coastal polynyas, dense overflows, and the sharp, v-shaped front of the western boundary current which Gordon et al. (2020) argued was crucial for the pathway of AABW export.

Regardless of these shortcomings, we feel that the time-dependent water mass framework presented here is a useful tool for understanding AABW variability in this region. A very promising direction for future work would be to apply the same methodology to higher-resolution models, which presumably represent small scale processes with much greater fidelity. A recent study by Stewart (2021) showed that a 1/12° regional model of the Weddell Sea could resolve the impact of tides and eddies on cross-shelf transfer of buoyancy, resulting in a more realistic overturning circulation. High resolution global ocean climate models (e.g. Small et al. (2014); Morrison et al. (2016); Kiss et al. (2020)) are also capable of resolving these processes in better

detail and would be a promising tool for investigating AABW WMT. One challenge, however, in applying the WMT approach to these models is the computationally demanding nature of the WMT diagnostics, which require closed heat and salt budgets to begin with, and then layer on additional complex calculations.

We also note the limitations of the potential-density WMT framework used here. Our method is not capable of explicitly diagnosing WMT effects due to the nonlinear equation of state—cabbeling and thermobaricity—even those processes are

hypothesized to be important for AABW formation (Gill, 1973; Gordon et al., 2020). Alternative frameworks use neutral density (Jackett and McDougall, 1997; Iudicone et al., 2008b) or move to two-dimensional temperature / salinity water-mass coordinates (Evans et al., 2018), which can reveal subtleties in the transformation process that are inaccessible to the 1D density-based approach. We look forward to exploring these possibilities in future work.

*Code and data availability.*

WMT budgets can be found in the author's *GitHub repository*. The analysis-ready datasets used for calculating the budgets can be found in the cloud through *Pangeo's catalog*.

*Author contributions.*

Shanice Bailey and Ryan Abernathey conceptualized the research goals. Shanice Bailey conducted the analysis with help from Spencer Jones and Ryan Abernathey. Shanice Bailey prepared the manuscript with significant contributions from all coauthors.

Arnold Gordon and Xiaojun Yuan provided scientific input and guidance.

## B   Special issue statement

This manuscript is also being submitted for the Special Issue on the Weddell Sea hosted by the Southern Ocean Observing System Weddell Sea Dronning Maud Land Regional Working Group.





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
