# Peer review of "Water mass transformation variability in the Weddell Sea in Ocean Reanalyses"

_EGUsphere, 2022_

## Referee Comment (RC1)

**Water mass transformation variability in the Weddell Sea in Ocean Reanalyses**

**General comments**

The authors use three different ocean reanalysis products and apply a water-mass transformation framework to study the processes that affect the variability of AABW exported from the Weddell Sea. This work addresses a climatically relevant process that plays an important role in the global Meridional Overturning Circulation. To my knowledge, there have not been similar studies using ocean reanalysis and it will be useful for the community to know whether these products are simulating realistically AABW formation processes and export. Their main conclusion is that the reanalyses are not capable of consistently and faithfully capture processes involved in driving AABW production, export and variability.

The application of the WMT framework in the region is a novel idea, but I find that there must be further explanation within the manuscript of the methodology and definitions used for the study in order to interpret the results. Considerable improvements to the structure of the manuscript should also be made to facilitate reading.

**Specific comments**

**Major comments**

1. You conclude in the Discussion section that none of the reanalyses capture realistically AABW formation, export and variability due to their coarse resolution. This should be included in the abstract, before describing the loss of AABW found in SOSE, not at the end as a reason for no relationship with large-scale climate oscillations.

2. It is not clear how AABW is defined in this paper. In line 287 defines it based on TS boundaries, in line 319 it is all the waters denser than CDW (without clarifying which density this is), and in line 364 there are different density criteria for each reanalysis. Since the paper is focusing on AABW, its definition should be clearer to the reader, with supporting references if it is based on prior studies, or adequate justification if it is a new definition for the purposes of this work.

3. The study region is not defined. Defining the study region is vital for the reader to interpret the physical meaning behind inflows/outflows. For example, is the shelf region included? Is the northern boundary set by topography or an arbitrary latitude?

4. The Introduction is devoted mostly to a description of the global MOC. I suggest shifting to a more regional focus that is more relevant to the paper. For example, what is already known about water-mass transformations and inflows/outflows of AABW. Some works that could be useful on this regard are:

   - Couldrey, M. P., Jullion, L., Naveira Garabato, A. C., Rye, C., Herráiz-Borreguero, L., Brown, P. J., ... & Speer, K. L. (2013). Remotely induced warming of Antarctic Bottom Water in the eastern Weddell gyre. Geophysical Research Letters, 40(11), 2755-2760.
   - Meredith, M. P., Gordon, A. L., Naveira Garabato, A. C., Abrahamsen, E. P., Huber, B. A., Jullion, L., & Venables, H. J. (2011). Synchronous intensification and warming of Antarctic Bottom Water outflow from the Weddell Gyre. Geophysical Research Letters, 38(3).

- Jullion, L., Naveira Garabato, A. C., Meredith, M. P., Holland, P. R., Courtois, P., & King, B. A. (2013). Decadal freshening of the Antarctic Bottom Water exported from the Weddell Sea. Journal of Climate, 26(20), 8111-8125.

Also, there is a lack of references in the Introduction and Discussion sections, including but not limited to:

- Line 28: "... forming Antarctic Bottom Water (AABW) [reference]"
- Line 31: "The global impacts of this circulation system on biological productivity and carbon and heat uptake [reference] …"
- Lines 32 to 35.
- Line 40: ".... interior mixing [reference]"
- Line 46: "... cascading down continental boundaries [reference]"
- Line 55: " … storage of large amounts of carbon [reference]"
- Line 73: "Even if physical processes such as coastal polynyas are not always represented accurately in ocean reanalysis [reference] ..."

And finally, please double check the relevance of citations. For example:

- Line 24: Purkey and Johnson 2013, Vernet et al. 2019 are not relevant citations regarding the MOC's inter-hemispheric transport
- Line 463: Armitage et al. 2018 don't study AABW production/export

5. In the model assessment section, in order to make a fair comparison, please select the same time periods for model/observations. Your comparison could be biased because you are comparing, for example, bottom temperatures from ECCO for the period 1992 to 2015 with WOA for the period 1981 to 2010. This applies to the entire section.

6. It's not clear to me that bottom properties in the reanalysis products are similar to the observed. What does this mean for the reliability of AABW production and representation in reanalyses?

7. Lines 318-325: it is confusing the grouping of CDW/WSDW. Did you mean CDW/WDW? Warm Deep Water is the local term for CDW in the Weddell region, whereas Weddell Sea Deep Water is a different water mass altogether, with a different origin and characteristics.

8. In the same paragraph, you cite export values from Figure 5. To get export values, shouldn't you get the cumulative sum for the density range corresponding to your AABW definition? If I interpreted correctly, that minimum is the cumulative sum of export of **all** the lighter densities, and is in fact excluding export from denser waters. And how are you obtaining the transformation values afterwards?

9. Line 326: Kerr et al (2012) obtain this value from a transect at the tip of the Antarctic Peninsula. Your calculation encompasses outflows for the entire region, so I'm not sure it is valid comparison. This comparison is repeated in other places in the text.

10. Paragraph starting in line 330: you are describing Figure 7 and your conclusion is that both models agree with past studies. However, I look at the Figure and ECCO and SOSE look completely different. They show different water mass transformations (different signs for same density ranges). And again, how are you obtaining the precise values for transformation of bottom waters?

11. In a couple of occasions, you mention that the salinity driven WMT is due to brine rejection, with no influence of ice melt, runoff or precip (such as line 357). How do you verify that?

12. In line 363 you define CDW as the inflowing water and AABW as the outflowing waters. There are several studies that show that there is significant inflow of AABW into the region. Have you verified how sensitive your results are on this definition?
13. Also, if I interpret your figure correctly, the time tendency in SOSE is <0 year-round. Wouldn't this mean that there is a loss of volume throughout the year? This contradicts your statement in line 368.
14. Are the correlations between terms and the correlations between climate indices significant? With what confidence level?
15. In the Discussion section you mention discrepancies with mooring records. Please cite them and describe how your analysis is different.

**Minor comments (general)**

- Correct figure captions throughout the text. For example: add units in Figure 1 and 2, add description of density contours in Figure 3 and 4 as well as labels for panels.
- Please use same axis for different panels. For example, Figures 5 to 8. Figure 10 is ok since it would make visualizing difficult.
- Sections 4 to 5 read a bit confusing. A suggestion for improvement is to try to make paragraphs focused on just one topic, which could be, for instance, describing one figure. Avoid one-sentence paragraphs as in line 326 and 460. Last for paragraphs in Seasonal Climatology section could be organised better: you go from describing Figure 7 for all 3 models, to a paragraph each describing ECCO and SOSE in Figure 8 and 9, back to SODA in Figure 7 for the last paragraph.
- There is no need to completely describe a figure in the text since it is already in the captions. For example, from line 310 onwards you describe all components of Figure 6 and 7, when you could limit the paragraph to (for example): The time evolution of the cumulative water mass volume distribution is not completely balanced by the total inflow/outflows and mean transformations, indicating that the system is not in steady state balance but is subject to low frequency variability or model drift".
- Avoid subjective judgements. For example: line 359: "It is interesting to note"; line 447: "it gets worse, not better"; line 505: "take these correlations with a grain of salt".

**Minor comments (line based)**

- Line 13: add period where SOSE shows loss of AABW.
- Line 25: "... transported to the *northern hemisphere* …"
- Add 1000m isobath in Figures 1 and 2 to aid in visualization.
- Consider changing salinity colorbar range or colormap to better visualize the region's bottom salinity, Figure 2.
- Add delimiting lines of AABW according to working definitions in the TS diagrams, Figures 3 and 4; as well as in Figures 5 and 6.
- Change units in all figures from [x] to (x).
- Figure 9: change colorbar label from Sverdrup (m3/s) to Transport (Sv).
- Number figures in order of appearance in the text (Figure 9 is described before than Figures 6 and 7)

- Line 275: the region you are indicating is not beneath the Filchner-Ronne Ice Shelf (I think none of these reanalyses include the FRIS).
- Line 351: why do you refer to *psi* (inflow/outflow from the region) to overturning?
- Line 383: why tertiary and not secondary?
- Line 478: change wording. For example, "there is little agreement between reanalysis products regarding variability in seasonal to interannual timescales"
- There are several syntax errors throughout the text, including but not limited to:
  - Line 438: Fig. 2 is not the one you wanted to reference right?
  - Line 439: forgot parenthesis closure
  - Line 457: and instead of AND
  - Line 465: sea-ice concentration (SIC).
  - Line 522: starts with Purkey and Johnson (2012) and ends with (Hellmer et al 2011).
- Correct citations throughout the text. For example, line 549: (e.g. Small et al. (2014); Morrison et al. (2016); Kiss et al. (2020)) should be (e.g. Small et al. 2014; Morrison et al. 2016; Kiss et al. 2020). This is probably a latex typo, please check thoughout the text.
- Similar to the comment above, verify throughout the text the figures you are referencing. For example, in line 517 you reference Fig. 2, but I don't think that is the figure you wanted to refer to. If so, I don't see how in bottom salinity you can identify loss of AABW properties.
- Check text for unclosed parenthesis (I found several).

---

## Author Comment (AC1)

We sincerely thank the editor and the reviewers for their attention to our manuscript and their helpful suggestions. We synthesize the reviews and summarize the overall areas where our manuscript needs work in the following list:

- We will improve the citations of prior literature and the presentation of these references in the introduction and discussions.
- We will clarify our method for partitioning the transport into two layers and better explain the cumulative nature of the streamfunction minimum as a metric of transport.
- We will decompose the surface-salinity-driven transformation into a sea-ice driven component and a component from direct E-P-R (as done in Abernathey et al. 2016).

In addition, we will make many smaller changes based on the reviewers' helpful suggestions.

Below is a point-by-point response to each reviewer comment.

**RC1 (Anonymous Referee #1)**

**Major Comments**

1. You conclude in the Discussion section that none of the reanalyses capture realistically AABW formation, export and variability due to their coarse resolution. This should be included in the abstract, before describing the loss of AABW found in SOSE, not at the end as a reason for no relationship with large-scale climate oscillations.
   Agreed. Will include our finding that the reanalyses used did not realistically capture AABW formation, export and variability in the abstract, BEFORE mentioning the findings from each model.

2. It is not clear how AABW is defined in this paper. In line 287 defines it based on TS boundaries, in line 319 it is all the waters denser than CDW (without clarifying which density this is), and in line 364 there are different density criteria for each reanalysis. Since the paper is focusing on AABW, its definition should be clearer to the reader, with supporting references if it is based on prior studies, or adequate justification if it is a new definition for the purposes of this work.
   Agreed. Will clearly state that the definition of the AABW depends on each model's annual mean overturning streamfunction (when the line switches signs, essentially); and this single value shall represent the range between the densest value up to the value at the boundary of the overturning circulation demarcating deep and bottom water masses.

3. The study region is not defined. Defining the study region is vital for the reader to interpret the physical meaning behind inflows/outflows. For example, is the shelf region included? Is the northern boundary set by topography or an arbitrary latitude?
   `Agreed. Boundary will be explicitly defined by these`
   `coordinates [65°W, 30°E, 78°S, 62°S] (Will cite Gordon et`
   `al 2007, DOI: 10.1175/JCLI4046.1 for these boundaries)`
4. The Introduction is devoted mostly to a description of the global MOC. I suggest shifting to a more regional focus that is more relevant to the paper. For example, what is already known about water-mass transformations and inflows/outflows of AABW. Some works that could be useful on this regard are:
   `This comment is a very similar comment to Céline Huezé, so`
   `we will condense the introduction to focus more on the`
   `study region and consider incorporating these suggested`
   `works into the text.`
   a. Couldrey, M. P., Jullion, L., Naveira Garabato, A. C., Rye, C., Herráiz-Borreguero, L., Brown, P. J., ... & Speer, K. L. (2013). Remotely induced warming of Antarctic Bottom Water in the eastern Weddell gyre. Geophysical Research Letters, 40(11), 2755- 2760.
   b. Meredith, M. P., Gordon, A. L., Naveira Garabato, A. C., Abrahamsen, E. P., Huber, B. A., Jullion, L., & Venables, H. J. (2011). Synchronous intensification and warming of Antarctic Bottom Water outflow from the Weddell Gyre. Geophysical Research Letters, 38(3).
   c. Jullion, L., Naveira Garabato, A. C., Meredith, M. P., Holland, P. R., Courtois, P., & King, B. A. (2013). Decadal freshening of the Antarctic Bottom Water exported from the Weddell Sea. Journal of Climate, 26(20), 8111-8125.

   Also, there is a lack of references in the Introduction and Discussion sections, including but not limited to:
   d. Line 28: "... forming Antarctic Bottom Water (AABW) [reference]"
      `We will cite Talley 2013.`
   e. Line 31: "The global impacts of this circulation system on biological productivity and carbon and heat uptake [reference] ..."
      `We will cite Vernet et al 2019 and highlight`
      `circulation in the WG (study region) within the`
      `context of the MOC is worth studying.`
   f. Lines 32 to 35.
      `We will cite Talley 2013 and Vernet et al 2019.`
   g. Line 40: ".... interior mixing [reference]"
      `We will cite Nikurashin and Vallis (2011,2012)`
   h. Line 46: "... cascading down continental boundaries [reference]"

> Line 46 also comes from Williams, 2001. We will cite
> Williams once at the end of this sentence instead of
> the sentence prior in line 45.

    i. Line 55: " ... storage of large amounts of carbon [reference]"
> We will cite Ito et al 2015:
> https://agupubs.onlinelibrary.wiley.com/doi/full/10.10
> 02/2015GL064320.

    j. Line 73: "Even if physical processes such as coastal polynyas are not always represented accurately in ocean reanalysis [reference] ..."
> Even though it is within this study's own experience
> with such reanalyses, we will find a reference to this
> statement.

And finally, please double check the relevance of citations. For example:

    k. Line 24: Purkey and Johnson 2013, Vernet et al. 2019 are not relevant citations regarding the MOC's inter-hemispheric transport
> Instead we will cite Talley 2013.

    l. Line 463: Armitage et al. 2018 don't study AABW production/export
> We will remove Armitage et al 2018 from line 463.

5. In the model assessment section, in order to make a fair comparison, please select the same time periods for model/observations. Your comparison could be biased because you are comparing, for example, bottom temperatures from ECCO for the period 1992 to 2015 with WOA for the period 1981 to 2010. This applies to the entire section.
> This is a reasonable suggestion. However, not all of the
> reanalyses cover the climate normal period, so it is not
> possible. Thus we prefer to keep the comparison as is;
> however, we will add text noting the possible issues with
> this comparison.

6. It's not clear to me that bottom properties in the reanalysis products are similar to the observed. What does this mean for the reliability of AABW production and representation in reanalyses?
> No direct observational estimates of AABW production exist.
> But direct observations of bottom water production do. In
> this section, we are using bottom water properties as a
> metric (clearly imperfect) for overall model accuracy. None
> of the reanalysis products provides a formal quantification
> of estimate uncertainty, so we use the model standard
> deviation instead.

7. Lines 318-325: it is confusing the grouping of CDW/WSDW. Did you mean CDW/WDW? Warm Deep Water is the local term for CDW in the Weddell region,

whereas Weddell Sea Deep Water is a different water mass altogether, with a different origin and characteristics.

`That is a typo, thank you!`

8. In the same paragraph, you cite export values from Figure 5. To get export values, shouldn't you get the cumulative sum for the density range corresponding to your AABW definition? If I interpreted correctly, that minimum is the cumulative sum of export of all the lighter densities, and is in fact excluding export from denser waters. And how are you obtaining the transformation values afterwards?

```
The other reviewer had similar questions, so clearly we
need to improve the description of our method. Our budget
coarsely separates all water into two classes: bottom water
(denser than sigma^cross) and everything else (by volume,
mostly CDW and its regional variants; thus the name
"deep"). Due to volume conservation, in this construction,
the transport of bottom water across the basin boundary is
equal and opposite to the transport of deep water. The
dividing isopycnal sigma^cross is defined via the minimum
of the overturning streamfunction. So one single transport
value represents both the deep outflow and the
equal-and-opposite inflow.
```

9. Line 326: Kerr et al (2012) obtain this value from a transect at the tip of the Antarctic Peninsula. Your calculation encompasses outflows for the entire region, so I'm not sure it is a valid comparison. This comparison is repeated in other places in the text.

```
We acknowledge that the locations differ. However, Kerr
does attempt to capture the dominant outflow of the Weddell
sea AABW. We think it is important to attempt to compare
with observations somehow. In our revision, we will note
this caveat in the discussion of Kerr.
```

10. Paragraph starting in line 330: you are describing Figure 7 and your conclusion is that both models agree with past studies. However, I look at the Figure and ECCO and SOSE look completely different. They show different water mass transformations (different signs for the same density ranges). And again, how are you obtaining the precise values for transformation of bottom waters?

```
In our revision, we will modify this language to be more
nuanced, rather than just saying "agree". (I assume you're
meaning Fig 6.) What we mean is that the different
components of transformation in each model align
qualitatively with established understanding of how the
circulation works. Each component of transformation
(surface salt, surface temp, and mixing) has the correct
```

sign (i.e. mixing acts to lighten the wm, brine rejection acts to increase AABW, etc). However, it is not possible to make a *quantitative* comparison of individual components, since no such prior calculations have been published; that is part of the novelty of this study.

11. In a couple of occasions, you mention that the salinity driven WMT is due to brine rejection, with no influence of ice melt, runoff or precip (such as line 357). How do you verify that?

Similar comment to Céline's major comment and specific comment #30. Similar response here: in our revised manuscript, we will quantify the role of sea ice vs. evaporation / precipitation / runoff  using model diagnostics to decompose the surface salinity contribution to transformation.

12. In line 363 you define CDW as the inflowing water and AABW as the outflowing waters. There are several studies that show that there is significant inflow of AABW into the region. Have you verified how sensitive your results are on this definition?

Our transport values are the net sum of all transport across the boundary, including both inflow and outflow, for a given density range. We only choose the separatrix density based on the inflow / outflow criterion. We will clarify this in our revision.

13. Also, if I interpret your figure correctly, the time tendency in SOSE is <0 year-round. Wouldn't this mean that there is a loss of volume throughout the year? This contradicts your statement in line 368.

Yes, that is the correct interpretation of Fig. 7. We agree that the statement in line 368 needs to be revised. The seasonal cycle in SOSE includes a large negative offset due to the time-mean volume loss.

14. Are the correlations between terms and the correlations between climate indices significant? With what confidence level?

In our revised manuscript, we will include confidence levels for these reported correlation values.

15. In the Discussion section you mention discrepancies with mooring records. Please cite them and describe how your analysis is different.

Mooring records were cited in line 496 and mentioned again in line 536. In our revision, we will make these citations more prominent and collocated with relevant discussion in the text. Our analysis is very different from those papers because it focuses on reanalysis products, not direct

observations. 1:1 comparison between moorings and
reanalysis is difficult because of the highly local nature
of mooring observations.

**Minor Comments (general)**

1. Correct figure captions throughout the text. For example: add units in Figure 1
   and 2, add description of density contours in Figure 3 and 4 as well as labels for
   panels.
   Agreed. We will add units to Figure 1 and 2(˚C and psu,
   respectively) and to Figure 5 (kg m^-3); and will add
   description of sigma2 density contours in Figs 3 and 4.
2. Please use the same axis for different panels. For example, Figures 5 to 8.
   Figure 10 is ok since it would make visualizing difficult.
   We will not change - x axis label was left to be only at
   the bottom to make comparative visuals easier for the
   reader.
3. Sections 4 to 5 read a bit confusing. A suggestion for improvement is to try to
   make paragraphs focused on just one topic, which could be, for instance,
   describing one figure. Avoid one-sentence paragraphs as in line 326 and 460.
   Lastly, paragraphs in the Seasonal Climatology section could be organised
   better: you go from describing Figure 7 for all 3 models, to a paragraph each
   describing ECCO and SOSE in Figure 8 and 9, back to SODA in Figure 7 for the
   last paragraph.
   We will focus on each section to be clearer. The purpose of
   inserting a paragraph about ECCO and SOSE before going back
   to SODA in Fig 7 is because we are describing each model in
   the order displayed in Fig 7; and Figs 8 & 9 give a deeper
   look into ECCO and SOSE.
4. There is no need to completely describe a figure in the text since it is already in
   the captions. For example, from line 310 onwards you describe all components of
   Figure 6 and 7, when you could limit the paragraph to (for example): The time
   evolution of the cumulative water mass volume distribution is not completely
   balanced by the total inflow/outflows and mean transformations, indicating that
   the system is not in steady state balance but is subject to low frequency
   variability or model drift".
   Will get rid of text from 310-316 and follow suggestions.

5. Avoid subjective judgements. For example: line 359: "It is interesting to note"; line 447: "it gets worse, not better"; line 505: "take these correlations with a grain of salt".
   `Will adjust the statements in line 447 to: "...and this trend persists…"; and line 505 to: "However, SOSE is very short, so these correlations are not definitive."`

**Minor Comments (line based)**

1. Line 13: add period where SOSE shows loss of AABW.
   `"...loss of AABW during 2005-2010, driven…"`
2. Line 25: "... transported to the northern hemisphere ..."
   `Will add suggestion to line.`
3. Add 1000m isobath in Figures 1 and 2 to aid in visualization.
   `Yes we will do this in our revision.`
4. Consider changing salinity colorbar range or colormap to better visualize the region's bottom salinity, Figure 2.
   `We will use a better colormap in our revision.`
5. Add delimiting lines of AABW according to working definitions in the TS diagrams, Figures 3 and 4; as well as in Figures 5 and 6.
   `We will have this range highlighted with a box in figs 3 and 4,and we will add delimiting lines of AABW in Figs 5 and 6.`
6. Change units in all figures from [x] to (x).
   `Will change unit brackets.`
7. Figure 9: change colorbar label from Sverdrup (m3/s) to Transport (Sv).
   `Will change label to Transport (Sv).`
8. Number figures in order of appearance in the text (Figure 9 is described before than Figures 6 and 7)
   `Will reorder the text to mention Figure 9 after explaining the layout of the figures 7 & 8 to be similar to that of Figs 5 & 6.`
9. Line 275: the region you are indicating is not beneath the Filchner-Ronne Ice Shelf (I think none of these reanalyses include the FRIS).
   `Thank you for catching that. We will adjust the text accordingly since the ice shelf ends more south than where the salt plumes are appearing.`
10. Line 351: why do you refer to psi (inflow/outflow from the region) to overturning?
    `Psi quantifies the strength of the overturning.`
11. Line 383: why tertiary and not secondary?

> Because its magnitude is less than the mixing component of
> transformation.

12. Line 478: change wording. For example, "there is little agreement between reanalysis products regarding variability in seasonal to interannual timescales"
> Will change to: "...there is little agreement between these
> reanalysis products on interannual timescales".

13. There are several syntax errors throughout the text, including but not limited to:
> Will fix syntax errors.

   a. Line 438: Fig. 2 is not the one you wanted to reference right?
   > Fig. 10!

   b. Line 439: forgot parenthesis closure
   > Will close parenthesis.

   c. Line 457: and instead of AND
   > Will change to "and".

   d. Line 465: sea-ice concentration (SIC).
   > Change abbreviation to "(SIC)".

   e. Line 522: starts with Purkey and Johnson (2012) and ends with (Hellmer et al 2011).
   > "In Hellmer et al. (2011), it is…"

14. Correct citations throughout the text. For example, line 549: (e.g. Small et al (2014); Morrison et al. (2016); Kiss et al. (2020)) should be (e.g. Small et al. 2014; Morrison et al. 2016; Kiss et al. 2020). This is probably a latex typo, please check throughout the text.
> We will correct citations where multiple papers are cited.

15. Similar to the comment above, verify throughout the text the figures you are referencing. For example, in line 517 you reference Fig. 2, but I don't think that is the figure you wanted to refer to. If so, I don't see how in bottom salinity you can identify loss of AABW properties.
> We meant Fig. 5; we will double check each fig referenced
> in text.

16. Check text for unclosed parenthesis (I found several).
> Will do, thank you!

**RC2 (Céline Heuzé)**

**General Comments**

My first issue is your definition of polynyas. [Disclaimer: The following contains references to papers from my team. I am not mentioning them to pressure you into

citing me. These are the studies I know of that illustrate the point I am about to make.] On several occasions in the text, you discuss how some signals look like polynya signatures. I would add that the surface salinity flux of SOSE on Fig 9 suggests this further. The issue I have is that you rule out polynyas based on sea ice concentration. Globally, and especially so over Maud Rise, there can be polynyas with a near-100% sea ice concentration… but with a very thin ice. That's why many polynya detection algorithms use a thickness threshold instead of a concentration one. A threshold of 12 cm is standard; see e.g. the latest Nakata et al. (doi: 10.1029/2020GL091353) for coastal polynyas, and Mohrmann et al. 2021 (doi:10.5194/tc-15-4281-2021) for all types of Antarctic polynyas, in models and in observations. That is particularly crucial because there was some halo / small polynya activity in the region in 2004 and 2005 (table B1 of Heuzé et al. 2021, doi: 10.5194/tc-15-3401-2021). I would therefore like to see you redo your "polynya analysis" with the sea ice thickness instead of the concentration, and if the thickness falls below the threshold or simply decreases, rewrite your discussions accordingly. An extra (supplementary?) figure to compare the sea ice in all three models would be most welcome. If they assimilate sea ice instead, do describe the data source, frequency, whether they even assimilate thickness or only concentration, etc.

We did preliminary "polynya analysis" with ECCO's sea ice
thickness diagnostic and defined the threshold as suggested
(0.12m). We found no link between the second type of polynya
(below the threshold thickness) and the AABW density contour
outcropping between 2004-2007. Below we have attached the first
year of the outcropping (2004). Similar patterns are observed
for the latter years.

[Figure]

We will conduct a similar analysis with SOSE's diagnostic for
the year 2005 to see if it agrees with Huezé et al. 2021's
finding. Also, to address this major comment and a specific one
further down (#30), we will examine the role of sea ice vs.
evaporation / precipitation / runoff  using model diagnostics to
decompose the surface salinity contribution to transformation.

My second issue is from line 365 onwards: I do not understand why you are conducting the analysis on sigma cross, the boundary between CDW and AABW, rather than on AABW itself. Especially since you say line 366 that you are using this level to study AABW. I would like to see a clearer explanation of why this particular level can be representative of AABW, and not, as I first expected, be where the signal is most dampened. I would also like to see that this choice is robust, either by providing a supplementary version of Figs 7, 8, 10 and 11 created at a denser level, or by adding "denser" lines to these figures.

The key point here is that, because the streamfunction is a *cumulative* integral quantity (eg. 8), we are not conducting an analysis "on sigma cross". This framework defines AABW transport as the flow across the boundary of *all water denser than sigma cross*. Choosing a denser value would only damp the signal we are interested in, since the transport values decline monotonically as density increases from sigma^cross (fig. 5). This was clearly not explained well in our manuscript, and in our revision we will clarify this important point.

**Specific Comments**

1. The introduction up till line 45 is quite repetitive. I would merge these paragraphs and keep only the key points, notably the very last line (first time you do not mention only air-sea interactions but also cryosphere-sea interactions).
   Will condense introduction to focus more on the study region.

2. The paragraph line 53-59 is out of place. That should be among the very first things to write about, this overall "why should anyone who is not an oceanographer care?". By this point, I would rather you explain why the seasonal to interannual variabilities are important to study, which processes they impact, etc.
   Will consider integrating this paragraph to the one below or eliminate it altogether.

3. Line 89, for context, provide the depth range that you are looking at in this study (should also be mentioned in the introduction)
   Our study uses density coordinates and consequently does not specifically address a depth range. In our revision, we will note the approximate depth ranges associated with our water masses.

4. Line 91-93: Discuss whether these limitations are problematic to study the Weddell Gyre region. Again, no pressure for citations, but Mohrmann et al. 2022 (doi:10.1029/2022GL098036) suspects that some mixing signals we see in observations in the region are the result of cabbeling and thermobaricity.

   `Thank you for pointing out this interesting new study. We will include this in the discussion in our revised manuscript. We note that we are not necessarily `*`neglecting`*` these thermodynamic processes; we are simply not explicitly diagnosing their contribution to WMT. They are lumped into the mixing terms.`

5. Line 103: typo I suspect, vertical is diapycnal? Also, since you called this term G_h, I would write "horizontal" instead of "lateral" in the description, to help the readers.

   `We will change "lateral" to "horizontal", and "isopycnal" to "diapycnal" for G_vdiff term.`

6. Having the reanalyses introduced earlier would make section 2.1 easier to picture, to know which variables are available at which resolution, esp. vertical. Consider whether to swap sections 2 and 3.

   `While we appreciate this suggestion, we prefer to lead with the theory if that is okay.`

7. Line 136 onwards: your cumulative sums, bottom up or surface down?

   `We use a bottom-up implementation. However, we also note that all results are insensitive to this choice, provided care is taken with the signs. The theoretical foundation for the cumsum is in eqs. 6 and 8, via the Heaviside function. One can always change the convention by multiplying everything by -1.`

8. Equation 14: double minus = typo?

   `Typo indeed! We will fix it.`

9. Section 3: for all products, specify whether sigma_2 is provided or whether you had to compute it, and if so, how.

   `Sigma_2 was computed using a python package called JMD95. When introducing sigma2, we will say sigma2 was computed using the Jackett & McDougall (1995) ocean equation of state and will `[cite]` accordingly.`

10. Line 194, ECCO: Vertical grid type? Resolution (in m/dbar)?

    `In meters.`

11. Throughout the manuscript, for example lines 205 and 207, check your citation styles (citet vs citep, if using LaTeX)

    `We will recheck every citation and fix the style accordingly.`

12. Line 2012, SOSE: Number of vertical levels? Type of vertical grid? Vertical resolution? Also, please write more clearly what daily values of 5-day averages mean: each day is the 5-day mean centred on that day?
    ```
    The model was constructed in spherical coordinates with 42
    vertical levels of varying depth (m). 5-day averaging
    starting 1-5 January 2005 (the centering window is
    unclear). We will add the info to text.
    ```

13. Line 226, SODA: Vertical grid type? Vertical resolution?
    ```
    The vertical level is in z^star coordinate. Vertical levels
    are at telescoping depths.
    ```

14. Line 232: how many ensemble members?
    ```
    Not explicitly told (DOI: 10.1175/JCLI-D-18-0149.1); just
    states, "growing ensemble".
    ```

15. Line 245-246: I do not understand this description; it is nudged? If so, how often? Towards which variable(s)?
    ```
    Not sure how often, but from what we came across for the
    budget analysis the salt field looked to be nudged.
    ```

16. Figure 1 / line 258 onwards: is the bottom at the same depth for all products? In particular, is the shelf-break shifted N-S?
    ```
    The bottom is the same depth for the three products.
    ```

17. Lines 262-263: I do not understand what the standard deviation and spatial variability mean. Throughout the manuscript, use consistent terms such as "temporal standard deviation", "horizontal standard deviation", etc
    ```
    We agree the terminology is unclear. In this case, we are
    referring to the temporal standard deviation, which we use
    as a proxy for the (unknown) reanalysis uncertainty. In our
    revision, we will use consistent terminology.
    ```

18. Figure 1: fonts are too small on the colorbars
    ```
    We will make cbar fonts larger.
    ```

19. Figure 2: fonts are too small there as well, and adjust the caxis ranges on most panels, but especially the first column, so that we see more details
    ```
    We will make cbar fonts larger, and consider changing the
    cbar range.
    ```

20. Line 288: Help the reader by having this range highlighted with a box on Fig3.
    ```
    Good suggestion. We will implement this in our revision.
    ```

21. Line 291: you eventually give this information several pages later, but give the T-S range of CDW now or on line 288, and not just that of AABW.
    ```
    Might disregard the density range for CDW altogether in the
    later sections.
    ```

22. Figure 4 deserves way longer a description than this short line 294. Comment on the overestimation (?) in CDW, or on how the Gade line really sticks out, which suggests some sea ice / ice shelf interaction misrepresentation.

    `Good suggestion. We will do that.`

23. Figure 3: something is off with SODA at T=0 deg C. Explain why it has a larger volume than expected, or correct if that is artificial.

    `Will look into it (might be another missing data (with`
    `wrong time date) that we missed to omit that's throwing it`
    `off).`

24. Figure 4: The caption and titles are reversed, so write more clearly "model minus WOA" (or the opposite)

    `We will change the caption and title for Fig 4 according to`
    `suggestion.`

25. Line 318: you eventually give the densities in the different models many pages from here. You should give them here instead. Because you just finished showing how biased they are in T-S, yet here it looks like you do not account for their biases in density.

    `We do not account for the biases when defining each model's`
    `density, we define the lowest density value of AABW for`
    `each model based on the overturning streamfunction in Fig 5`
    `(i.e. on the boundary between deep and bottom water).`

26. Line 325: what does a volume gain mean in practice? Takes over other water masses higher up in the water column? Wider branch?

    `Volume gain means other water masses are transformed into`
    `AABW - they add to the volume of the AABW mass class.`

27. Line 331: Fig 6 does not show brine rejection, only surface salinity fluxes. Could be P-E. This joins my major comment: your study of sea ice needs to be more extensive, and to be shown.

    `We are currently working on breaking down the`
    `salinity-driven transformation into a component based on`
    `sea ice and a component based on direct E-P-R, as was done`
    `by Abernathey et al. (2016). These results will be included`
    `in the revised manuscript.`

28. Line 336-337: are these small values a typo? If not, which density interval am I supposed to look at right now? Because I did not notice this order of magnitude difference from ECCO.

    `These values are not a typo. The values of transformation`
    `components are computed on the boundary between deep and`
    `bottom water masses. As per previous suggestions, we will`
    `put delineating lines to denote the boundary between deep`

and bottom so that it may be easier for the reader to discern these values in the figure.

29. Line 347: I guess you do not use LaTeX after all. Please do not comment on Fig 9 before Figs 7 and 8, it is quite uncomfortable to have to go back and forth between the pages.
We did use LaTex to prepare the manuscript. In our revision, we will make sure to present and discuss the figures in the correct order.

30. Line 358: see major comment, surface cooling + freshening suggests sea ice melt (from below) to me.
Please see the response under the major comment.

31. Line 420-421: have you tried contacting them? They may not have checked AABW, but maybe they've investigated NADW for the AMOC and that would give you clues.
We tried contacting them on a different matter, but received no response.

32. Figure 10: to help with the comparison of the variabilities, have the same width as time interval for the three panels (e.g. 1 cm per year). Ideally, align them even, so that we can directly compare the models to each other.
For our revision, we will make this figure and see how it looks. Our concern is that the SOSE time period is very short compared to the others.

33. Figure 11: same comment as Fig 10
Same response as above.

34. Line 541: there's an Heuzé 2021 on CMIP6 (doi: 5194/os-17-59-2021). More models, more recent, same conclusions. As previously: no expectation of citation, just for your information.
We will include. Thank you!

---

## Referee Report (RR1)

**Water mass transformation variability in the Weddell Sea in Ocean Reanalyses – 2nd submission**

The authors have replied to the comments raised during the reviews, but have not addressed thoroughly all the concerns, some of which constitute major weak points of the work. The application of the WMT framework to AABW in the region is a novel aspect of this study, is interesting and worth doing. However, I still have major concerns on some aspects of the study and its methodology, which is why I don't recommend its acceptance and publication at this stage. I encourage the authors to do further work and resubmit at a future stage.

In particular, my main suggestion is that given that one of the main conclusions is that **the reanalysis are not capturing observed variability, and that the reanalyses are not consistently or faithfully capturing the processes that drive AABW in a robust way (line 556),** and that **the valuable aspect of the work is the WMT transformation framework developed**; why not do this analysis with a high resolution ocean model that is able to capture AABW more robustly? See for example Solodoch et al 2022 (using Kiss et al 2020 model which you mention in the conclusion). This would probably provide more insightful results because I see many issues with reanalyses throughout the manuscript, which removes relevance from your main findings listed in the abstract (volume loss in SOSE, larger interannual variability in ECCO, unphysically large variability in SODA).

**Comments not addressed in the response**

Below I expand further on the responses to the review that I consider did not properly address the concerns. The numbering corresponds to the comment's number in the Author's Response document.

**3. and 9.** The definition of the boundary region in this study is key to interpreting the results of the analysis (accounting for part of the volume changes according to equation 7 in the manuscript), and not enough work has been done to corroborate that the results are robust to the choice of boundary. The definition based on latitude/longitude lines seems arbitrary, based on a study that focuses on the Weddell Polynya in the vicinity of Maud Rise, rather than on other studies more relevant to the topic at hand (such as Solodoch et al 2022, that has with a description of the AABW pathways of export from the region). And the interpretation of export across these boundaries throughout the paper is also misleading. For example:

- Line 9: in the abstract it says "*we diagnose a closed form of the water mass budget for AABW that explicitly accounts for transport across the WG boundary*". The lat/lon lines chosen actually cut through the Weddell Gyre (see for example Neme et al 2021), and therefore the export does not represent export through the gyre's boundaries. It might even be capturing recirculations within the gyre itself.
- Line 342: "*We note that though Kerr et al. (2012) obtained this value from a transect at the tip of the Antarctic Peninsula, they do attempt to capture the dominant outflow of the Weddell sea AABW.*" Again, this is not a fair comparison: you are encompassing other circulation pathways and recirculations within the gyre in your export calculation

because of your boundary definition, whereas Kerr et al (2012) looked at transport across a specific hydrographic transect. You have responded to comment 7 from the review document "*We think it is important to attempt to compare with observations somehow.*" If comparison to observations is what you were after, given that the reanalyses are gridded products, you could have reproduced the hydrographic transect, which would have made for a better comparison against observations.

The above dot points are examples of how the selection of boundaries influence the results and conclusions of this paper. I don't consider there is appropriate justification behind this selection.

**5.** This comment regarding Figures 1 and 2 has not been addressed. I understand that not all the reanalyses are available during the same period, but as your Figure 10 shows, there is significant interannual variability that is going to influence this comparison. In other words, you are not comparing apples to apples here. You have added to the text "*We still compare each model to the available observation, but recognize that this introduces unknown biases in our comparison.*", but it is important to highlight that these needn't be *unknown biases*. It should be easy and straightforward to do this comparison with the same time period to observe the biases, and even if afterwards you decide to keep your Figures 1 and 2 the same, at least then you have proper justification for the reviewer and the reader, as well as understanding about how your decision is impacting your results.

**10.** I still think that SOSE is showing that most of the transformations of your definition of AABW is accounted by surface heat fluxes, whereas in ECCO it is mostly due to surface salt fluxes (which you call brine rejection, see following comment), which are qualitatively different. In fact, ECCO shows almost no transformation due to surface cooling.

**11.** There is still no justification to the transformation attributed to brine rejection. The orange dashed line in your Figure 6 is showing surface salinity flux, which does not necessarily represent brine rejection. In line 346 you say that your analysis shows a negligible role from the atmosphere, but you don't show the analysis or describe it? In your response to the reviews it says you were going to do a decomposition a la Abernathey et al 2016, but I can't see any reference to that in the revised version.

**Other comments**

1. The Introduction is still not well organised: you start by (i) a description of the MOC, follow with (ii) a very brief paragraph on the Weddell Gyre, (iii) then the relevance of AABW for the global climate, (iv) then what obs. show regarding AABW export, (v) then WMT framework, and end with (vi) your research question.
   A clearer view of your research area, in my opinion, would be to start with (iii) the relevance of AABW, part of which is its connection to (i) the MOC. You could then follow with a description of the characteristics and processes linked to AABW production/circulation/export in your study region, which would encompass its production mediated by the (ii) Weddell Gyre, as well as (iv) the observational studies. This would provide an adequate frame and highlight that we *don't know the thermodynamic mechanisms that link variability in surface forcing to AABW export*, and

describe (v) the WMT framework, which is your methodology to study your (vi) research questions (thermodynamic mechanisms).

2. **This is a major comment stemming from additions to the manuscript. You have used Jackett and McDougall (1995) equation of state to define sigma2. Why have you not used the most recent equation of state (TEOS10)? There is even a python library called gws that calculates sigma2 easily.**

3. You have made some changes to the Figures that have not improved them: you have used the Blue to Reds colorbar for anomalies, which is standard procedure and its perfect; but then for the standard deviation you switch from a sequential colorbar in Fig. 1 to the Blue-Red in Fig. 2 which at a glance could be misleading for the reader. And I would have also suggested to use a different colorbar for bottom temperatures, because again at a glance they could read as anomalies.

4. You use standard deviation in Figures 1 and 2 as a *proxy of the (unknown) reanalyses uncertainties* (line 271). This is not correct since standard deviation is a measure of variability.

5. Line 330: you say that because of volume conservation, the transport of bottom water is equal and opposite to the transport of deep water. Is this not affected by surface mass fluxes?

6. Line 374: "excess dense water is created by WMT in winter and then destroyed in summer". This is not the case for ECCO, where there is production throughout the year (green line panel a.)

7. **Line 439: the residual is your ECCO time series in Figure 10a is large, with the same order of magnitude of your time series of transport, WMT and change in volume. You say you find no explanation in this, are you sure your calculation is not flawed? This is worrying because for your interannual time series you conclude that SODA's is flawed, SOSE is to short and therefore ECCO is the most useful, even though it presents these large errors.**

8. There are still minor typos (which I won't go into detail because it is not the focus of this review) and not-so-minor errors throughout the text that makes me think it was not proofread. These include, but are not limited to: there is a random sentence in line 247 that seems should not be there, acronyms not defined in the text (NADW line 57), one sentence paragraphs (line 121 and 170), equations that are referenced before they are defined (line 111)

References

Neme, J., England, M. H., & Hogg, A. M. (2021). Seasonal and Interannual Variability of the Weddell Gyre From a High-Resolution Global Ocean-Sea Ice Simulation During 1958–2018. *Journal of Geophysical Research: Oceans*, *126*(11), e2021JC017662.

Solodoch, A., Stewart, A. L., Hogg, A. M., Morrison, A. K., Kiss, A. E., Thompson, A. F., ... & Cimoli, L. (2022). How Does Antarctic Bottom Water Cross the Southern Ocean?. *Geophysical Research Letters*, *49*(7), e2021GL097211.

---

## Author Response (AR4)

**Editor Comments - small abstract edit**

One small but in my view important suggestion was not completely carried out in the second last version and which slipped my attention, and I like to ask you to still follow up: You have addressed my earlier comment, namely:

"I like to suggest you shift the focus of the objective, and reflect/stress that in the intro , discussion and abstract, of the study slightly by calling your work an evaluation of the reanalyses to obtain WMT estimates and variability, and not to actually investigate the (Ed. 'real or observed') variability of WMT."

... in the Introduction and Discussion, however, this has not been done in the abstract. Please do so by simpling including a couple of words in the first sentences in the abstract.

Changed one of the sentences in the abstract to more accurately reflect the shift of the focus of the paper: "*This study explores how current state of the art data-assimilating ocean reanalyses can help fill the gaps in our understanding of the thermodynamic drivers of AABW variability in the WG via WMT volume budgets derived from Walin's classic WMT framework. The three ocean reanalyses used are:...*"

**Editor Comments - third round**

We sincerely thank the editor for accepting our manuscript subject to the minor revisions specified below!

The title page of *pdf. manuscript file must include the full institutional addresses of all authors. However, country name is missing from the affiliations. Please add it for the next revision.

Country name, United states of America, has been added to every address line on the title page. Thank you!

Red means final response.

We sincerely thank the editor and R1 for their attention to our manuscript and their very helpful suggestions. Here is a summary of what was addressed during this round:
- We reran the calculations with a more northern boundary defined for the WG, on par with the northern boundary value used in Neme et al 2021 (57˚S). This resulted in results that make more sense (e.g. lower influence from residuals, consistent relationship between

transformation and export in ECCO's interannual variability), and highlights the robustness of the WMT framework! So we want to again thank R1 for pointing out our initial boundary definition cutting over key dynamics. We note that this was a very significant amount of work. We essentially had to redo our entire analysis from the beginning.

- As suggested by the editor we slightly shifted the focus to say that we explored how these ocean reanalyses behaved in the WMT framework.
- We have followed R1's suggestion in reordering the introduction
- We have responded to a major comment about running the calculations using ACCESS-OM2
- We have also changed the color schemes for all of the figures, except for the TS histogram figs, so as to comply with Copernicus' request to make figures accessible to readers with color vision deficiencies.

Below is a point-by-point response to each reviewer comment.

**Editor Comments - second round**

I like to suggest you shift the focus of the objective, and reflect/stress that in the intro , discussion and abstract, of the study slightly by calling your work an evaluation of the reanalyses to obtain WMT estimates and variability, and not to actually investigate the variability of WMT. I believe R1 does have several points still that need to be clarified, dealt with and finetuned, however, I do not support the view of R1 that this study should rather be carried out by a high resolution ocean model instead - as is suggested in the 2nd paragraph of the report #1. A way of dealing with this, is to present the results as an evaluation/comparison of the reanalyses to capture WMT and AAWB, rather than an investigation of the variability of AABW itself (given the conclusions that they do not capture it too well). I hope you are able to present your revised ms with a slight shift of its focus, and address the other specific points and main comments left by R1 still.

We have shifted the wording in our focus of this study. Please see line 70 in the introduction, 552 in the Discussion section. We think this highlights the point that we used these models as tools to try and understand the internal thermodynamics happening within the WG; however, we still acknowledge that these are not ground truths, but simply the best option we currently have in studying such a remote region where in situ observations are sparse.

**RC1 - second round**

The authors have replied to the comments raised during the reviews, but have not addressed thoroughly all the concerns, some of which constitute major weak points of the work. The application of the WMT framework to AABW in the region is a novel aspect of this study, is interesting and worth doing. However, I still have major concerns on some aspects of the study and its methodology, which is why I don't recommend its acceptance and publication at this stage. I encourage the authors to do further work and resubmit at a future stage.

We thank the reviewer for taking the time to raise these concerns and hope that our substantial revisions, including new analysis, are sufficient for publication.

In particular, my main suggestion is that given that one of the main conclusions is that the reanalysis are not capturing observed variability, and that the reanalyses are not consistently or faithfully capturing the processes that drive AABW in a robust way (line 556), and that the valuable aspect of the work is the WMT transformation framework developed; why not do this analysis with a high resolution ocean model that is able to capture AABW more robustly? See for example Solodoch et al 2022 (using Kiss et al 2020 model which you mention in the conclusion). This would probably provide more insightful results because I see many issues with reanalyses throughout the manuscript, which removes relevance from your main findings listed in the abstract (volume loss in SOSE, larger interannual variability in ECCO, unphysically large variability in SODA).

While there are clearly shortcomings in these data-assimilating models, the independent observational evidence of water mass transformation and circulation in the Weddell Sea is extremely sparse. We don't think it's possible to say conclusively that these models are "not capturing observed variability"; there are no observations of the interannual variability in the basin-wide WMT or overturning circulation, only isolated moorings. Data assimilating models do their best to fit all available observations with a consistent physical solution. We maintain that exploring their solutions is a worthwhile contribution, despite potential model shortcomings. We never claim that these timeseries represent the true history of WMT in the Weddell Sea; only that they represent plausible, thermodynamically consistent mechanisms of interannual variability.

In our revised analysis, which includes a larger region better aligned with the physical boundaries of the Weddell Gyre, the volume trends on SOSE are much less pronounced and the interannual variability in ECCO is reduced. This mitigates some of the reviewer concerns.

We think applying the WMT framework to a high-resolution model
such as ACCESS-OM2 would be a great target for a future study.
In addition to resolution, the primary difference is that the
models used in our analysis are all data-assimilating models,
meaning that the model runs are constrained by observational
data. ACCESS-OM2 uses GFDL's MOM6 model which is unconstrained
by observations and belongs to a separate class of ocean models.

**Comments not addressed in response**

The numbering corresponds to the comment's number in the Author's Response document.

**3. and 9.** The definition of the boundary region in this study is key to interpreting the results of the analysis (accounting for part of the volume changes according to equation 7 in the manuscript), and not enough work has been done to corroborate that the results are robust to the choice of boundary. The definition based on latitude/longitude lines seems arbitrary, based on a study that focuses on the Weddell Polynya in the vicinity of Maud Rise, rather than on other studies more relevant to the topic at hand (such as Solodoch et al 2022, that has with a description of the AABW pathways of export from the region). And the interpretation of export across these boundaries throughout the paper is also misleading. For example:

- Line 9: in the abstract it says "*we diagnose a closed form of the water mass budget for AABW that explicitly accounts for transport across the WG boundary*". The lat/lon lines chosen actually cut through the Weddell Gyre (see for example Neme et al 2021), and therefore the export does not represent export through the gyre's boundaries. It might even be capturing recirculations within the gyre itself.
- Line 342: "*We note that though Kerr et al. (2012) obtained this value from a transect at the tip of the Antarctic Peninsula, they do attempt to capture the dominant outflow of the Weddell sea AABW.*" Again, this is not a fair comparison: you are encompassing other circulation pathways and recirculations within the gyre in your export calculation because of your boundary definition, whereas Kerr et al (2012) looked at transport across a specific hydrographic transect. You have responded to comment 7 from the review document "*We think it is important to attempt to compare with observations somehow.*" If comparison to observations is what you were after, given that the reanalyses are gridded products, you could have reproduced the hydrographic transect, which would have made for a better comparison against observations.

The above dot points are examples of how the selection of boundaries influence the results and conclusions of this paper. I don't consider there is appropriate justification behind this selection.

```
We have rerun all calculations using an extended northern
boundary (from 62˚S to 57˚S).
```

**5.** This comment regarding Figures 1 and 2 has not been addressed. I understand that not all the reanalyses are available during the same period, but as your Figure 10 shows, there is significant interannual variability that is going to influence this comparison. In other words, you are not comparing apples to apples here. You have added to the text "*We still compare each model to the available observation, but recognize that this introduces unknown biases in our comparison.*", but it is important to highlight that these needn't be *unknown biases*. It should be easy and straightforward to do this comparison with the same time period to observe the biases, and even if afterwards you decide to keep your Figures 1 and 2 the same, at least then you have proper justification for the reviewer and the reader, as well as understanding about how your decision is impacting your results.

```
We appreciate this comment. Based on this recommendation, we
have switched out the observational product to another WOA
product that covers 2005-2012. This is the closest in-situ data
we could find of bottom temp/salt fields in the WG to SOSE's
time period (2005-2010). As noted in the revised paper, ECCO and
SODA have also been sliced and averaged over WOA's time span.
```

**10.** I still think that SOSE is showing that most of the transformations of your definition of AABW is accounted by surface heat fluxes, whereas in ECCO it is mostly due to surface salt fluxes (which you call brine rejection, see following comment), which are qualitatively different. In fact, ECCO shows almost no transformation due to surface cooling.

```
With the new figure 6 reflecting the result of an extended
northern boundary we see that transformation in ECCO is mainly
driven by salt fluxes; whereas in SOSE it is driven by mixing
(and is countered equally by surface salt and heat fluxes).
Qualitatively, both models still agree with the literature that
surface salt and heat fluxes act to increase AABW volume and
mixing acts to homogenize/destroy AABW.
```
**11.** There is still no justification to the transformation attributed to brine rejection. The orange dashed line in your Figure 6 is showing surface salinity flux, which does not necessarily represent brine rejection. In line 346 you say that your analysis shows a

negligible role from the atmosphere, but you don't show the analysis or describe it? In your response to the reviews it says you were going to do a decomposition a la Abernathey et al 2016, but I can't see any reference to that in the revised version.

Below is a figure of the influences from sea ice and atmosphere on surface salt fluxes. The black line is the total freshwater flux. The red dashed line represents the freshwater flux from the atmosphere to the ocean, while the blue dashed line represents the freshwater flux from sea ice to the ocean. The gray dashed line represents the linear free surface term to correct for a rigid lid which serves as a residual due to model construction. As you can see between the red and blue lines, the influence of sea-ice-derived freshwater fluxes are 5x larger than that from the atmosphere. We are further able to decompose the freshwater flux from sea ice to the ocean by categorizing negative fluxes to represent brine rejection/sea ice formation and positive fluxes to represent ice melt.

[Figure]

**Other comments**

1. The Introduction is still not well organised: you start by (i) a description of the MOC, follow with (ii) a very brief paragraph on the Weddell Gyre, (iii) then the relevance of AABW for the global climate, (iv) then what obs. show regarding AABW export, (v) then WMT framework, and end with (vi) your research question. A clearer view of your research area, in my opinion, would be to start with (iii) the relevance of AABW, part of which is its connection to (i) the MOC. You could then follow with a description of the characteristics and processes linked to AABW production/circulation/export in your study region, which would encompass its production mediated by the (ii) Weddell Gyre, as well as (iv) the observational studies. This would provide an adequate frame and highlight that we *don't know the thermodynamic mechanisms that link variability in surface forcing to AABW export*, and describe (v) the WMT framework, which is your methodology to study your (vi) research question (thermodynamic mechanisms).

   ```
   Very good suggestion, thank you! We have implemented this
   in our revision (see lines 1-76).
   ```

2. **This is a major comment stemming from additions to the manuscript. You have used Jackett and McDougall (1995) equation of state to define sigma2. Why have you not used the most recent equation of state (TEOS10)? There is even a python library called gws that calculates sigma2 easily.**

   ```
   The ECCO and SOSE models are products of the MITgcm and
   uses the JMD95 version of EOS; therefore, to be consistent
   we use this version of EOS to define sigma2 for all 3
   models. Using a different EOS than the model's own internal
   EOS would lead to spurious results.
   ```

3. You have made some changes to the Figures that have not improved them: you have used the Blue to Reds colorbar for anomalies, which is standard procedure and its perfect; but then for the standard deviation you switch from a sequential colorbar in Fig. 1 to the Blue-Red in Fig. 2 which at a glance could be misleading for the reader. And I would have also suggested to use a different colorbar for bottom temperatures, because again at a glance they could read as anomalies.

   ```
   Adjusted bottom temperature and salt fields by using
   cmocean's thermal and haline colormaps. Changed salt's
   standard deviation plots to be consistent with
   temperature's standard deviation plots. Kept difference
   plots the same Red-Blues.
   ```

4. You use standard deviation in Figures 1 and 2 as a proxy of the (unknown) reanalyses uncertainties (line 271). This is not correct since standard deviation is a measure of variability.

```
We understand this concern but this is the best we can do
given the lack of reported uncertainties with the model
data.
```

5. **Line 330:** you say that because of volume conservation, the transport of bottom water is equal and opposite to the transport of deep water. Is this not affected by surface mass fluxes?

```
The surface mass fluxes are several orders of magnitude
smaller than the overturning and WMT and so for simplicity
have not been included in the budgets.
```

6. Line 374: "excess dense water is created by WMT in winter and then destroyed in summer". This is not the case for ECCO, where there is production throughout the year (green line panel a.)

```
Thank you for pointing that out. We will make the text more
clear and accurate by adding to the line: "...; however, in
ECCO the destruction is caused by less contribution from
transformation and the influence of outflow becomes
compensatory."
```

7. **Line 439: the residual is your ECCO time series in Figure 10a is large, with the same order of magnitude of your time series of transport, WMT and change in volume. You say you find no explanation in this, are you sure your calculation is not flawed? This is worrying because for your interannual time series you conclude that SODA's is flawed, SOSE is to short and therefore ECCO is the most useful, even though it presents these large errors**

```
Great question. New study region (more northern boundary)
produced a figure with R1 being very low in magnitude
compared to the other budget terms!
```

8. There are still minor typos (which I won't go into detail because it is not the focus of this review) and not-so-minor errors throughout the text that makes me think it was not proofread. These include, but are not limited to: there is a random sentence in line 247 that seems should not be there, acronyms not defined in the text (NADW line 57), one sentence paragraphs (line 121 and 170), equations that are referenced before they are defined (line 111)

References

Neme, J., England, M. H., & Hogg, A. M. (2021). Seasonal and Interannual Variability of the Weddell Gyre From a High-Resolution Global Ocean-Sea Ice Simulation During 1958– 2018. Journal of Geophysical Research: Oceans, 126(11), e2021JC017662.

Solodoch, A., Stewart, A. L., Hogg, A. M., Morrison, A. K., Kiss, A. E., Thompson, A. F., ... & Cimoli, L. (2022). How Does Antarctic Bottom Water Cross the Southern
Ocean?. Geophysical Research Letters, 49(7), e2021GL097211.

Blue means comments have been addressed

We sincerely thank the editor and the reviewers for their attention to our manuscript and their helpful suggestions. We synthesize the reviews and summarize the overall areas where our manuscript needs work in the following list:
- We will improve the citations of prior literature and the presentation of these references in the introduction and discussions.
- We will clarify our method for partitioning the transport into two layers and better explain the cumulative nature of the streamfunction minimum as a metric of transport.
- We will decompose the surface-salinity-driven transformation into a sea-ice driven component and a component from direct E-P-R (as done in Abernathey et al. 2016).

In addition, we will make many smaller changes based on the reviewers' helpful suggestions.

Below is a point-by-point response to each reviewer comment.

**RC1 (Anonymous Referee #1)**

**Major Comments**

1. You conclude in the Discussion section that none of the reanalyses capture realistically AABW formation, export and variability due to their coarse resolution. This should be included in the abstract, before describing the loss of AABW found in SOSE, not at the end as a reason for no relationship with large-scale climate oscillations.

   ```
   Agreed. Will include our finding that the reanalyses used
   did not realistically capture AABW formation, export and
   variability in the abstract, BEFORE mentioning the findings
   from each model.
   ```

2. It is not clear how AABW is defined in this paper. In line 287 defines it based on TS boundaries, in line 319 it is all the waters denser than CDW (without clarifying which density this is), and in line 364 there are different density criteria for each reanalysis. Since the paper is focusing on AABW, its definition should be clearer

to the reader, with supporting references if it is based on prior studies, or adequate justification if it is a new definition for the purposes of this work.

```
Agreed. Will clearly state that the definition of the AABW
depends on each model's annual mean overturning
streamfunction (when the line switches signs, essentially);
and this single value shall represent the range between the
densest value up to the value at the boundary of the
overturning circulation demarcating deep and bottom water
masses.
```

3. The study region is not defined. Defining the study region is vital for the reader to interpret the physical meaning behind inflows/outflows. For example, is the shelf region included? Is the northern boundary set by topography or an arbitrary latitude?

```
Agreed. Boundary will be explicitly defined by these
coordinates [65°W, 30°E, 78°S, 62°S] (Will cite Gordon et
al 2007, DOI: 10.1175/JCLI4046.1 for these boundaries)
```

4. The Introduction is devoted mostly to a description of the global MOC. I suggest shifting to a more regional focus that is more relevant to the paper. For example, what is already known about water-mass transformations and inflows/outflows of AABW. Some works that could be useful on this regard are:

```
This comment is a very similar comment to Céline Huezé, so
we will condense the introduction to focus more on the
study region and consider incorporating these suggested
works into the text.
```

    a. Couldrey, M. P., Jullion, L., Naveira Garabato, A. C., Rye, C., Herráiz-Borreguero, L., Brown, P. J., ... & Speer, K. L. (2013). Remotely induced warming of Antarctic Bottom Water in the eastern Weddell gyre. Geophysical Research Letters, 40(11), 2755- 2760.

    b. Meredith, M. P., Gordon, A. L., Naveira Garabato, A. C., Abrahamsen, E. P., Huber, B. A., Jullion, L., & Venables, H. J. (2011). Synchronous intensification and warming of Antarctic Bottom Water outflow from the Weddell Gyre. Geophysical Research Letters, 38(3).

    c. Jullion, L., Naveira Garabato, A. C., Meredith, M. P., Holland, P. R., Courtois, P., & King, B. A. (2013). Decadal freshening of the Antarctic Bottom Water exported from the Weddell Sea. Journal of Climate, 26(20), 8111-8125.

```
We have already cited this work in the discussion.
```

Also, there is a lack of references in the Introduction and Discussion sections, including but not limited to:

    d. Line 28: "... forming Antarctic Bottom Water (AABW) [reference]"

```
We will cite Talley 2013.
```

```
We will cite Vernet et al 2019 and highlight
circulation in the WG (study region) within the
context of the MOC is worth studying.
```

```
We will cite Talley 2013 Gordon 2020 and Vernet et al
2019.
```

```
We will cite Nikurashin and Vallis (2011,2012)
```

```
Line 46 also comes from Williams, 2001. We will cite
Williams once at the end of this sentence instead of
the sentence prior in line 45.
```

```
We will cite Ito et al 2015:
https://agupubs.onlinelibrary.wiley.com/doi/full/10.10
02/2015GL064320.
```

```
Even though it is within this study's own experience
with such reanalyses, we will find a reference to this
statement.
```
And finally, please double check the relevance of citations. For example:

```
Instead we will cite Talley 2013.
```

```
We will remove Armitage et al 2018 from line 463.
```

5. In the model assessment section, in order to make a fair comparison, please select the same time periods for model/observations. Your comparison could be biased because you are comparing, for example, bottom temperatures from ECCO for the period 1992 to 2015 with WOA for the period 1981 to 2010. This applies to the entire section.
```
This is a reasonable suggestion. However, not all of the
reanalyses cover the climate normal period, so it is not
possible. Thus we prefer to keep the comparison as is;
however, we will add text noting the possible issues with
this comparison.
```

6. It's not clear to me that bottom properties in the reanalysis products are similar to the observed. What does this mean for the reliability of AABW production and representation in reanalyses?

   ```
   No direct observational estimates of AABW production exist.
   But direct observations of bottom water production do. In
   this section, we are using bottom water properties as a
   metric (clearly imperfect) for overall model accuracy. None
   of the reanalysis products provides a formal quantification
   of estimate uncertainty, so we use the model standard
   deviation instead.
   ```

7. Lines 318-325: it is confusing the grouping of CDW/WSDW. Did you mean CDW/WDW? Warm Deep Water is the local term for CDW in the Weddell region, whereas Weddell Sea Deep Water is a different water mass altogether, with a different origin and characteristics.

   ```
   That is a typo, thank you!
   ```

8. In the same paragraph, you cite export values from Figure 5. To get export values, shouldn't you get the cumulative sum for the density range corresponding to your AABW definition? If I interpreted correctly, that minimum is the cumulative sum of export of all the lighter densities, and is in fact excluding export from denser waters. And how are you obtaining the transformation values afterwards?

   ```
   The other reviewer had similar questions, so clearly we
   need to improve the description of our method. Our budget
   coarsely separates all water into two classes: bottom water
   (denser than sigma^cross) and everything else (by volume,
   mostly CDW and its regional variants; thus the name
   "deep"). Due to volume conservation, in this construction,
   the transport of bottom water across the basin boundary is
   equal and opposite to the transport of deep water. The
   dividing isopycnal sigma^cross is defined via the minimum
   of the overturning streamfunction. So one single transport
   value represents both the deep outflow and the
   equal-and-opposite inflow.
   ```

9. Line 326: Kerr et al (2012) obtain this value from a transect at the tip of the Antarctic Peninsula. Your calculation encompasses outflows for the entire region, so I'm not sure it is a valid comparison. This comparison is repeated in other places in the text.

   ```
   We acknowledge that the locations differ. However, Kerr
   does attempt to capture the dominant outflow of the Weddell
   sea AABW. We think it is important to attempt to compare
   with observations somehow. In our revision, we will note
   this caveat in the discussion of Kerr.
   ```

10. Paragraph starting in line 330: you are describing Figure 7 and your conclusion is that both models agree with past studies. However, I look at the Figure and ECCO and SOSE look completely different. They show different water mass transformations (different signs for the same density ranges). And again, how are you obtaining the precise values for transformation of bottom waters?

   In our revision, we will modify this language to be more nuanced, rather than just saying "agree". (I assume you're meaning Fig 6.) What we mean is that the different components of transformation in each model align qualitatively with established understanding of how the circulation works. Each component of transformation (surface salt, surface temp, and mixing) has the correct sign (i.e. mixing acts to lighten the wm, brine rejection acts to increase AABW, etc). However, it is not possible to make a *quantitative* comparison of individual components, since no such prior calculations have been published; that is part of the novelty of this study.

11. In a couple of occasions, you mention that the salinity driven WMT is due to brine rejection, with no influence of ice melt, runoff or precip (such as line 357). How do you verify that?

   Similar comment to Céline's major comment and specific comment #30. Similar response here: in our revised manuscript, we will quantify the role of sea ice vs. evaporation / precipitation / runoff  using model diagnostics to decompose the surface salinity contribution to transformation.

12. In line 363 you define CDW as the inflowing water and AABW as the outflowing waters. There are several studies that show that there is significant inflow of AABW into the region. Have you verified how sensitive your results are on this definition?

   Our transport values are the net sum of all transport across the boundary, including both inflow and outflow, for a given density range. We only choose the separatrix density based on the inflow / outflow criterion. We will clarify this in our revision.

13. Also, if I interpret your figure correctly, the time tendency in SOSE is <0 year-round. Wouldn't this mean that there is a loss of volume throughout the year? This contradicts your statement in line 368.

   Yes, that is the correct interpretation of Fig. 7. We agree that the statement in line 368 needs to be revised. The

```
seasonal cycle in SOSE includes a large negative offset due
to the time-mean volume loss.
```

14. Are the correlations between terms and the correlations between climate indices significant? With what confidence level?

```
In our revised manuscript, we will include confidence
levels for these reported correlation values.
Update: updated fig 12 with error bars and included p
values and confidence intervals as well.
```

15. In the Discussion section you mention discrepancies with mooring records. Please cite them and describe how your analysis is different.

```
Mooring records were cited in line 496 and mentioned again
in line 536. In our revision, we will make these citations
more prominent and collocated with relevant discussion in
the text. Our analysis is very different from those papers
because it focuses on reanalysis products, not direct
observations. 1:1 comparison between moorings and
reanalysis is difficult because of the highly local nature
of mooring observations.
```

**Minor Comments (general)**

1. Correct figure captions throughout the text. For example: add units in Figure 1 and 2, add description of density contours in Figure 3 and 4 as well as labels for panels.

```
Agreed. We will add units to Figure 1 and 2(˚C and psu,
respectively) and to Figure 5 (kg m^-3); and will add
description of sigma2 density contours in Figs 3 and 4.
```

2. Please use the same axis for different panels. For example, Figures 5 to 8. Figure 10 is ok since it would make visualizing difficult.

```
We will not change - x axis label was left to be only at
the bottom to make comparative visuals easier for the
reader.
```

3. Sections 4 to 5 read a bit confusing. A suggestion for improvement is to try to make paragraphs focused on just one topic, which could be, for instance, describing one figure. Avoid one-sentence paragraphs as in line 326 and 460. Lastly, paragraphs in the Seasonal Climatology section could be organised better: you go from describing Figure 7 for all 3 models, to a paragraph each describing ECCO and SOSE in Figure 8 and 9, back to SODA in Figure 7 for the last paragraph.

We will focus on each section to be clearer. The purpose of
inserting a paragraph about ECCO and SOSE before going back
to SODA in Fig 7 is because we are describing each model in
the order displayed in Fig 7; and Figs 8 & 9 give a deeper
look into ECCO and SOSE.
**Update:** Did not change text order of Sects 4.1 and 5. The
lead author found the analysis in the text in a good order
for the reader to digest the figures. Changed order of text
in Sect. 4.2.

4. There is no need to completely describe a figure in the text since it is already in
   the captions. For example, from line 310 onwards you describe all components of
   Figure 6 and 7, when you could limit the paragraph to (for example): The time
   evolution of the cumulative water mass volume distribution is not completely
   balanced by the total inflow/outflows and mean transformations, indicating that
   the system is not in steady state balance but is subject to low frequency
   variability or model drift".
   Will get rid of text from 310-316 and follow suggestions.

5. Avoid subjective judgements. For example: line 359: "It is interesting to note"; line
   447: "it gets worse, not better"; line 505: "take these correlations with a grain of
   salt".
   Will adjust the statements in line 447 to: "...and this
   trend  amplifies…"; and line 505 to: "However, SOSE
   is very short, so these correlations are not definitive."

**Minor Comments (line based)**

1. Line 13: add period where SOSE shows loss of AABW.
   "...loss of AABW during 2005-2010, driven…"
2. Line 25: "... transported to the northern hemisphere ..."
   Will add suggestion to line.
3. Add 1000m isobath in Figures 1 and 2 to aid in visualization.
   Yes we will do this in our revision.
4. Consider changing salinity colorbar range or colormap to better visualize the
   region's bottom salinity, Figure 2.
   We will use a better colormap in our revision.
5. Add delimiting lines of AABW according to working definitions in the TS
   diagrams, Figures 3 and 4; as well as in Figures 5 and 6.
   We will have this range highlighted with a box in figs 3
   and 4,and we will add delimiting lines of AABW in Figs 5
   and 6.

6. Change units in all figures from [x] to (x).
   ```
   Will change unit brackets.
   ```
7. Figure 9: change colorbar label from Sverdrup (m3/s) to Transport (Sv).
   ```
   Will change label to Transport (Sv).
   ```
8. Number figures in order of appearance in the text (Figure 9 is described before than Figures 6 and 7)
   ```
   Will reorder the text to mention Figure 9 after explaining
   the layout of the figures 7 & 8 to be similar to that of
   Figs 5 & 6.
   ```
   ```
   Update: Instead we just switched the order of figs 8 & 9
   and kept the mentions in the text the same.
   ```
9. Line 275: the region you are indicating is not beneath the Filchner-Ronne Ice Shelf (I think none of these reanalyses include the FRIS).
   ```
   Thank you for catching that. We will adjust the text
   accordingly since the ice shelf ends more south than where
   the salt plumes are appearing.
   ```
10. Line 351: why do you refer to psi (inflow/outflow from the region) to overturning?
    ```
    Psi quantifies the strength of the overturning.
    ```
11. Line 383: why tertiary and not secondary?
    ```
    Because its magnitude is less than the mixing component of
    transformation.
    ```
12. Line 478: change wording. For example, "there is little agreement between reanalysis products regarding variability in seasonal to interannual timescales"
    ```
    Will change to: "...there is little agreement between these
    reanalysis products on interannual timescales".
    ```
13. There are several syntax errors throughout the text, including but not limited to:
    ```
    Will fix syntax errors.
    ```
    a. Line 438: Fig. 2 is not the one you wanted to reference right?
       ```
       Fig. 5!
       ```
    b. Line 439: forgot parenthesis closure
       ```
       Will close parenthesis.
       ```
    c. Line 457: and instead of AND
       ```
       Will change to "and".
       ```
    d. Line 465: sea-ice concentration (SIC).
       ```
       Change abbreviation to "(SIC)".
       ```
    e. Line 522: starts with Purkey and Johnson (2012) and ends with (Hellmer et al 2011).
       ```
       "In Hellmer et al. (2011), it is…"
       ```
14. Correct citations throughout the text. For example, line 549: (e.g. Small et al (2014); Morrison et al. (2016); Kiss et al. (2020)) should be (e.g. Small et al.

2014; Morrison et al. 2016; Kiss et al. 2020). This is probably a latex typo, please check throughout the text.

```
We will correct citations where multiple papers are cited.
```

15. Similar to the comment above, verify throughout the text the figures you are referencing. For example, in line 517 you reference Fig. 2, but I don't think that is the figure you wanted to refer to. If so, I don't see how in bottom salinity you can identify loss of AABW properties.

```
We meant Fig. 5; we will double check each fig referenced
in text.
```

16. Check text for unclosed parenthesis (I found several).

```
Will do, thank you!
```

**RC2 (Céline Heuzé)**

**General Comments**

My first issue is your definition of polynyas. [Disclaimer: The following contains references to papers from my team. I am not mentioning them to pressure you into citing me. These are the studies I know of that illustrate the point I am about to make.] On several occasions in the text, you discuss how some signals look like polynya signatures. I would add that the surface salinity flux of SOSE on Fig 9 suggests this further. The issue I have is that you rule out polynyas based on sea ice concentration. Globally, and especially so over Maud Rise, there can be polynyas with a near-100% sea ice concentration… but with a very thin ice. That's why many polynya detection algorithms use a thickness threshold instead of a concentration one. A threshold of 12 cm is standard; see e.g. the latest Nakata et al. (doi: 10.1029/2020GL091353) for coastal polynyas, and Mohrmann et al. 2021 (doi:10.5194/tc-15-4281-2021) for all types of Antarctic polynyas, in models and in observations. That is particularly crucial because there was some halo / small polynya activity in the region in 2004 and 2005 (table B1 of Heuzé et al. 2021, doi: 10.5194/tc-15-3401-2021). I would therefore like to see you redo your "polynya analysis" with the sea ice thickness instead of the concentration, and if the thickness falls below the threshold or simply decreases, rewrite your discussions accordingly. An extra (supplementary?) figure to compare the sea ice in all three models would be most welcome. If they assimilate sea ice instead, do describe the data source, frequency, whether they even assimilate thickness or only concentration, etc.

```
We did preliminary "polynya analysis" with ECCO's sea ice
thickness diagnostic and defined the threshold as suggested
(0.12m). We found no link between the second type of polynya
(below the threshold thickness) and the AABW density contour
```

outcropping between 2004-2007. Below we have attached the first
year of the outcropping (2004). Similar patterns are observed
for the latter years.

[Figure]

We will conduct a similar analysis with SOSE's diagnostic for
the year 2005 to see if it agrees with Huezé et al. 2021's
finding. Also, to address this major comment and a specific one
further down (#30), we will examine the role of sea ice vs.
evaporation / precipitation / runoff  using model diagnostics to
decompose the surface salinity contribution to transformation.
**Update:** We conducted similar analysis with SOSE and found
minimal contribution of E-P-R to surface salt fluxes, and
dominant brine rejection activity throughout most of the season
until summer when there were spikes in ice melt during a few
summer months (amplitude usually matching the highest brine
rejection spike).The new manuscript includes several
descriptions of this new analysis in relevant locations
(beginning in lines 164 and 346, for polynya analysis line 445).
We also found no link between the second type of polynya and the
AABW density contour outcropping. These findings were included
in the relevant sections where the text talks about possible
polynya activity and considering sea ice as the only influence
on transformation due to the surface salt fluxes.

My second issue is from line 365 onwards: I do not understand why you are conducting
the analysis on sigma cross, the boundary between CDW and AABW, rather than on
AABW itself. Especially since you say line 366 that you are using this level to study
AABW. I would like to see a clearer explanation of why this particular level can be
representative of AABW, and not, as I first expected, be where the signal is most
dampened. I would also like to see that this choice is robust, either by providing a
supplementary version of Figs 7, 8, 10 and 11 created at a denser level, or by adding
"denser" lines to these figures.

The key point here is that, because the streamfunction is a *cumulative* integral quantity (eg. 8), we are not conducting an analysis "on sigma cross". This framework defines AABW transport as the flow across the boundary of *all water denser than sigma cross*. Choosing a denser value would only damp the signal we are interested in, since the transport values decline monotonically as density increases from sigma^cross (fig. 5). This was clearly not explained well in our manuscript, and in our revision we will clarify this important point.

**Specific Comments**

1. The introduction up till line 45 is quite repetitive. I would merge these paragraphs and keep only the key points, notably the very last line (first time you do not mention only air-sea interactions but also cryosphere-sea interactions).
   Will condense introduction to focus more on the study region.

2. The paragraph line 53-59 is out of place. That should be among the very first things to write about, this overall "why should anyone who is not an oceanographer care?". By this point, I would rather you explain why the seasonal to interannual variabilities are important to study, which processes they impact, etc.
   Will consider integrating this paragraph to the one below or eliminate it altogether.

3. Line 89, for context, provide the depth range that you are looking at in this study (should also be mentioned in the introduction)
   Our study uses density coordinates and consequently does not specifically address a depth range. In our revision, we will note the approximate depth ranges associated with our water masses.

4. Line 91-93: Discuss whether these limitations are problematic to study the Weddell Gyre region. Again, no pressure for citations, but Mohrmann et al. 2022 (doi:10.1029/2022GL098036) suspects that some mixing signals we see in observations in the region are the result of cabbeling and thermobaricity.
   Thank you for pointing out this interesting new study. We will include this in the discussion in our revised manuscript. We note that we are not necessarily *neglecting* these thermodynamic processes; we are simply not explicitly diagnosing their contribution to WMT. They are lumped into the mixing terms.

5. Line 103: typo I suspect, vertical is diapycnal? Also, since you called this term G_h, I would write "horizontal" instead of "lateral" in the description, to help the readers.

   We will change "lateral" to "horizontal", and "isopycnal" to "diapycnal" for G_vdiff term.

6. Having the reanalyses introduced earlier would make section 2.1 easier to picture, to know which variables are available at which resolution, esp. vertical. Consider whether to swap sections 2 and 3.

   While we appreciate this suggestion, we prefer to lead with the theory if that is okay.

7. Line 136 onwards: your cumulative sums, bottom up or surface down?

   We use a bottom-up implementation. However, we also note that all results are insensitive to this choice, provided care is taken with the signs. The theoretical foundation for the cumsum is in eqs. 6 and 8, via the Heaviside function. One can always change the convention by multiplying everything by -1.

8. Equation 14: double minus = typo?

   Typo indeed! We will fix it.

9. Section 3: for all products, specify whether sigma_2 is provided or whether you had to compute it, and if so, how.

   Sigma_2 was computed using a python package called JMD95. When introducing sigma2, we will say sigma2 was computed using the Jackett & McDougall (1995) ocean equation of state and will cite accordingly.

10. Line 194, ECCO: Vertical grid type? Resolution (in m/dbar)?

    In meters.

11. Throughout the manuscript, for example lines 205 and 207, check your citation styles (citet vs citep, if using LaTeX)

    We will recheck every citation and fix the style accordingly.

12. Line 2012, SOSE: Number of vertical levels? Type of vertical grid? Vertical resolution? Also, please write more clearly what daily values of 5-day averages mean: each day is the 5-day mean centred on that day?

    The model was constructed in spherical coordinates with 42 vertical levels of varying depth (m). 5-day averaging starting 1-5 January 2005 (the centering window is unclear). We will add the info to text.

13. Line 226, SODA: Vertical grid type? Vertical resolution?

    The vertical level is in z^star coordinate. Vertical levels are at telescoping depths.

14. Line 232: how many ensemble members?
    Not explicitly told (DOI: 10.1175/JCLI-D-18-0149.1); just
    states, "growing ensemble".
15. Line 245-246: I do not understand this description; it is nudged? If so, how often?
    Towards which variable(s)?
    Not sure how often, but from what we came across for the
    budget analysis the salt field looked to be nudged.
16. Figure 1 / line 258 onwards: is the bottom at the same depth for all products? In
    particular, is the shelf-break shifted N-S?
    The bottom is the same depth for the three products.
17. Lines 262-263: I do not understand what the standard deviation and spatial
    variability mean. Throughout the manuscript, use consistent terms such as
    "temporal standard deviation", "horizontal standard deviation", etc
    We agree the terminology is unclear. In this case, we are
    referring to the temporal standard deviation, which we use
    as a proxy for the (unknown) reanalysis uncertainty. In our
    revision, we will use consistent terminology.
18. Figure 1: fonts are too small on the colorbars
    We will make cbar fonts larger.
19. Figure 2: fonts are too small there as well, and adjust the caxis ranges on most
    panels, but especially the first column, so that we see more details
    We will make cbar fonts larger, and consider changing the
    cbar range.
20. Line 288: Help the reader by having this range highlighted with a box on Fig3.
    Good suggestion. We will implement this in our revision.
21. Line 291: you eventually give this information several pages later, but give the
    T-S range of CDW now or on line 288, and not just that of AABW.
    Might disregard the density range for CDW altogether in the
    later sections.
    **Update:** We never provided the TS range for CDW.
22. Figure 4 deserves way longer a description than this short line 294. Comment on
    the overestimation (?) in CDW, or on how the Gade line really sticks out, which
    suggests some sea ice / ice shelf interaction misrepresentation.
    Good suggestion. We will do that.
    **Update:** We commented on the CDW distribution in Fig 4. We
    also computed and plotted the Gade line in Figs 3 & 4 but
    struggled to see the misrepresentation you suggested;
    therefore, we did not feel comfortable including it in the
    paper. Please see the aforementioned Gade line in the figs
    below.

[Figure]

TS Diagrams

23. Figure 3: something is off with SODA at T=0 deg C. Explain why it has a larger volume than expected, or correct if that is artificial.
    Will look into it (might be another missing data (with wrong time date) that we missed to omit that's throwing it off).

**Update:** Added text to talk about how we cannot explain this
feature.

24. Figure 4: The caption and titles are reversed, so write more clearly "model minus WOA" (or the opposite)

   We will change the caption and title for Fig 4 according to
   suggestion.

25. Line 318: you eventually give the densities in the different models many pages from here. You should give them here instead. Because you just finished showing how biased they are in T-S, yet here it looks like you do not account for their biases in density.

   We do not account for the biases when defining each model's
   density, we define the lowest density value of AABW for
   each model based on the overturning streamfunction in Fig 5
   (i.e. on the boundary between deep and bottom water).

26. Line 325: what does a volume gain mean in practice? Takes over other water masses higher up in the water column? Wider branch?

   Volume gain means other water masses are transformed into
   AABW - they add to the volume of the AABW mass class.

27. Line 331: Fig 6 does not show brine rejection, only surface salinity fluxes. Could be P-E. This joins my major comment: your study of sea ice needs to be more extensive, and to be shown.

   We are currently working on breaking down the
   salinity-driven transformation into a component based on
   sea ice and a component based on direct E-P-R, as was done
   by Abernathey et al. (2016). These results will be included
   in the revised manuscript.

28. Line 336-337: are these small values a typo? If not, which density interval am I supposed to look at right now? Because I did not notice this order of magnitude difference from ECCO.

   These values are not a typo. The values of transformation
   components are computed on the boundary between deep and
   bottom water masses. As per previous suggestions, we will
   put delineating lines to denote the boundary between deep
   and bottom so that it may be easier for the reader to
   discern these values in the figure.

29. Line 347: I guess you do not use LaTeX after all. Please do not comment on Fig 9 before Figs 7 and 8, it is quite uncomfortable to have to go back and forth between the pages.

   We did use LaTex to prepare the manuscript. In our
   revision, we will make sure to present and discuss the
   figures in the correct order.

30. Line 358: see major comment, surface cooling + freshening suggests sea ice melt (from below) to me.
    ```
    Please see the response under the major comment.
    ```
    **Update:** `changed 'corresponding to surface freshening' to 'corresponding to ice melt'.`

31. Line 420-421: have you tried contacting them? They may not have checked AABW, but maybe they've investigated NADW for the AMOC and that would give you clues.
    ```
    We tried contacting them on a different matter, but
    received no response.
    ```

32. Figure 10: to help with the comparison of the variabilities, have the same width as time interval for the three panels (e.g. 1 cm per year). Ideally, align them even, so that we can directly compare the models to each other.
    ```
    For our revision, we will make this figure and see how it
    looks. Our concern is that the SOSE time period is very
    short compared to the others.
    ```

33. Figure 11: same comment as Fig 10
    ```
    Same response as above.
    ```

34. Line 541: there's an Heuzé 2021 on CMIP6 (doi: 5194/os-17-59-2021). More models, more recent, same conclusions. As previously: no expectation of citation, just for your information.
    ```
    We will include. Thank you!
    ```